# Dynamic balance of myoplasmic energetics, redox state and protons in a fast-twitch oxidative glycolytic skeletal muscle fibre

Jana Disch[1] , Jeroen A.L. Jeneson[2,3] , Daniel A. Beard[4] , Oliver Röhrle[1,5] and Thomas Klotz[1]

[1] *Institute for Modelling and Simulation of Biomechanical Systems, University of Stuttgart, Stuttgart, Germany*
[2] *Center for Child Development and Exercise, Wilhelmina Children's Hospital, University Medical Center Utrecht, the Netherlands*
[3] *Biomedical MR Research Lab, Amsterdam University Medical Center|site AMC, the Netherlands*
[4] *Department of Molecular & Integrative Physiology, University of Michigan, USA*
[5] *Stuttgart Centre for Simulation Science (SC SimTech), University of Stuttgart, Germany*

Handling Editors: Bettina Mittendorfer & Christopher Sundberg

The peer review history is available in the Supporting Information section of this article (https://doi.org/10.1113/JP289702#support-information-section).

**Abstract figure legend** We have developed a computational model of energy metabolism in fast-twitch oxidative glycolytic muscle fibres. The model considers thermodynamically constrained enzyme kinetics derived from *in vitro* data and was validated against *in vivo* data from phosphorus magnetic resonance spectroscopy. This enables unique insights into energy, redox and proton balance in working skeletal muscle fibres.

**Jana Disch** has a background in Biomedical Engineering from the University of Tübingen and the University of Stuttgart. She is currently a doctoral candidate in the Department of Continuum Biomechanics and Mechanobiology at the Institute for Modelling and Simulation of Biomechanical Systems at the University of Stuttgart. Her research focuses on developing mathematical and computational frameworks for skeletal muscle energy metabolism to simulate and ultimately predict physiological function in health and disease.

**Abstract**  To investigate the mechanisms governing energy and redox balance in skeletal muscle, we developed a computational model describing the coupled biochemical reaction network of glycolysis and mitochondrial oxidative phosphorylation (OxPhos) in fast-twitch oxidative glycolytic (FOG) muscle fibres. The model was identified against dynamic *in vivo* recordings of phosphocreatine (PCr), inorganic phosphate (Pi) and pH in rodent hindlimb muscle and verified against independent data from *in vivo* experiments and muscle biopsies. Step response testing reveals that mass action kinetics in combination with feedback control are sufficient to accomplish myoplasmic ATP homeostasis over a 100-fold range of ATP turnover rates. This vital emergent property of the metabolic model is associated with intermediary metabolite dynamics typical of a second-order underdamped system, which has been previously reported for the glycolytic pathway. Lactate dehydrogenase (LDH) knockout simulations suggest that the contribution of the LDH reaction to redox balance is more fundamental to muscle function than its role in counteracting myoplasmic acidification across the physiological range of ATP demands in this myofibre phenotype. Furthermore, LDH knockout simulations confirm that mitochondrial uptake of myoplasmic NADH and $H^+$ in and by itself is sufficient to maintain redox balance and proton balance over ATP turnover rates in the range of mitochondrial ATP synthesis. We conclude that aerobic lactate production in working muscles is a by-product of the metabolic flexibility of FOG myofibres afforded by expression of high levels of LDH and OxPhos enzymes to support continual myoplasmic redox balance and ATP synthesis under conditions of high-intensity mechanical work.

(Received 11 July 2025; accepted after revision 12 December 2025; first published online 12 February 2026)

**Corresponding author** T. Klotz, Institute for Modelling and Simulation of Biomechanical Systems, University of Stuttgart, Stuttgart, Germany.  Email: thomas.klotz@imsb.uni-stuttgart.de

**Key points**

- Feedback regulation suffices to accomplish myoplasmic ATP homeostasis over a 100-fold range of ATP turnover.
- Second-order underdamped behaviour is predicted to arise as a generic trait of the ATP metabolic network in mammalian cells.
- Aerobic lactate is a by-product of the metabolic and functional flexibility.
- LDH's role in maintaining redox balance is more important than its role in counteracting cellular acidification.

## Introduction

Skeletal muscles are typically composed of a mix of fatigue-resistant and fatigue-prone myofibres, affording performance of a wide range of mechanical tasks. Each myofibre phenotype is made up of a specific set of protein isoforms in concentrations tailored to the fibre's particular function, ranging from slow-twitch (ST) oxidative 'red' fibres to fast-twitch (FT) glycogenolytic 'white' fibres (e.g. see MacIntosh et al., 2006; Schiaffino & Reggiani, 2011). Fast-twitch oxidative glycolytic (FOG) myofibres represent a hybrid phenotype that ranks both functionally and metabolically between these two fibre types.

The metabolic phenotypes of red and white myofibres represent opposite extremes, with red fibres reliant on the oxidation of fats and carbohydrates to fuel oxidative phosphorylation (OxPhos) and white fibres principally relying on the substrate-level phosphorylation of glycogen (MacIntosh et al., 2006; Schiaffino & Reggiani, 2011), respectively, to maintain ATP balance during contractile work. The capillary contact surface areas of each of these two extreme myofibre phenotypes support these metabolic traits (Glancy et al., 2014). The intermediate FOG myofibre phenotype, on the contrary, has both a high concentration of glycolytic enzymes, including LDH, and a relatively high mitochondrial density and capillary contact surface (Glancy et al., 2014). As a result, FOG fibres can operate under conditions of simultaneous pyruvate oxidation and lactate production through LDH. This ability endows FOG myofibres with metabolic flexibility to synthesise ATP under conditions of varying oxygen supply, such as may intermittently occur during muscular work, causing capillary collapse due to increased intramuscular pressure (Degens et al., 1998).

This hybrid oxidative–anaerobic ATP synthetic metabolic network in FOG myofibres may, however,

give rise to the so-called 'aerobic lactate production', whereby a fraction of pyruvate is converted to myoplasmic lactate despite ample residual oxidative capacity (Glancy et al., 2021; Sahlin et al., 1987). In terms of the efficiency of myoplasmic ATP synthesis, any aerobic lactate production is wasteful, as mitochondrial oxidation of NADH and pyruvate potentially yields far more ATP than glycolysis. However lactate transported to the extracellular milieu via any of the abundant organic anion exchangers in the myoplasmic membrane (e.g. MCT) (Juel & Halestrap, 1999) from one cell type does not necessarily represent a metabolic inefficiency at the level of an organism, as lactate is used to supply reducing equivalents and pyruvate for oxidative ATP synthesis in the heart and hepatic gluconeogenesis during exercise. Over the 100-fold operational range of ATP turnover rates that skeletal FT myofibres must absorb and balance (Glancy & Balaban, 2021), the mechanisms governing net pyruvate mass flow into the mitochondrial reticulum *versus* myoplasmic LDH are important not only to redox and energy balance in the myofibre but also in the whole body.

In resting muscle, the cytoplasmic NAD$^+$/NADH ratio has been estimated to be on the order of 400–900 (Glancy et al., 2021). Any production of myoplasmic NADH by GAPDH in the proximal part of glycolysis upon metabolic activation must be balanced by reoxidation of NAD$^+$ to NADH by myoplasmic–mitochondrial NADH shuttles and/or LDH (Glancy et al., 2021; Wang et al., 2022). Quantitative experimental methods to measure the dynamics of myoplasmic NAD(H) redox levels *in situ* are currently, however, missing. NADH fluorescence recordings from muscle, for example, are dominated by signal from mitochondrial matrix NADH (Hogan et al., 2005).

To further understand the dynamic balance of myoplasmic energetics, redox state and pH in an active FT muscle fibre, we used computer simulations to investigate ATP and pyruvate metabolism in FOG myofibres. First, we constructed a computational model of the coupled biochemical reaction network of glycolysis, LDH and mitochondrial oxidative ADP phosphorylation (OxPhos). The model was next identified for the FOG myofibre phenotype based on dynamic *in vivo* recordings of ATP metabolic levels and pH in rat FT muscle, and validated against independent data from dynamic *in vivo* recordings in FT muscles, as well as muscle biopsy analyses. Simulations then predicted that, on the one hand, mitochondrial pyruvate oxidation is the dominant source of ATP towards myoplasmic ATP balance over a 100-fold range of ATP turnover rates, but myoplasmic NADH conversion by LDH, on the other hand, structurally contributes to myoplasmic redox balance and surpasses oxidation of NADH by mitochondrial shuttles about midway through this range.

## Materials and methods

### Model

The model describes energy metabolism in contracting FOG fibres. The ATP turnover (Section 2.1.2) is balanced by ATP re-synthesis through (i) the buffer reactions catalysed by creatine kinase (CK) and adenylate kinase (AK) (Section 2.1.2), (ii) glyco(geno)lysis (Section 2.1.2) and (iii) OxPhos (Section 2.1.2). This yields a network of biochemical reactions described through thermodynamically constrained kinetic equations. A schematic overview of the proposed model is shown in Fig. 1. The metabolic network is located in a myoplasmic compartment. Separate mitochondrial domains within the myoplasm were not included. Two additional compartments represent the interstitial fluid and capillaries, respectively. For a detailed description of the model, see the Supplementary Material.

**Basic concept.** In this section, we provide a general description of the utilised modelling framework. For further details, see Alberty (2003); Vinnakota et al. (2006).

*Proton balance.* Most metabolites are (negatively charged) ions and, hence, exist in various species (bound to protons or metal ions), with rapid kinetic transitions that can be assumed to be in thermodynamic equilibrium. The total concentration of an arbitrary metabolite L is thus given by

$$[\mathrm{L_{tot}}] = [\mathrm{L}] + \sum_{p=1}^{N_\mathrm{P}} \left[\mathrm{H}_p\mathrm{L}\right] + \sum_{q=1}^{N_\mathrm{q}} \left[\mathrm{M}^q\mathrm{L}\right], \qquad (1)$$

with [L] representing the concentration of the unbound, most deprotonated species in a pH range of 5.5–8.5. Furthermore, $p$ denotes the stoichiometry number of bound protons, with $N_\mathrm{P}$ denoting the number of protonised species and $N_\mathrm{q}$ is the number of metal-bound species. As a simplification we assume that only one metal ion of each type can bind to L (with $N_\mathrm{q} = 2$ whereby $\mathrm{M}^1 = \mathrm{Mg}^{2+}$ and $\mathrm{M}^2 = \mathrm{K}^+$) and proton binding to metal-bound species is neglected (Vinnakota et al., 2006). The pH-dependent concentration of each species can be computed using the dissociation constants:

$$\left[\mathrm{H}_p\mathrm{L}\right] = \frac{[\mathrm{L}]\left[\mathrm{H}^+\right]^p}{\prod_{l=1}^{p} K_\mathrm{a}^l}, \; p \geqq 1, \qquad (2\mathrm{a})$$

$$\left[\mathrm{LM}^q\right] = \frac{[\mathrm{L}]\,[\mathrm{M}^q]}{K_\mathrm{d}^q}. \qquad (2\mathrm{b})$$

Therein $K_\mathrm{a}^l$ is the acid dissociation constant of the reaction $[\mathrm{H}_l\mathrm{L}] \rightleftarrows [\mathrm{H}_{l-1}\mathrm{L}] + \left[\mathrm{H}^+\right]$ and $K_\mathrm{d}^q$ the

dissociation constant for the binding reaction to the given metal ion type $M^q$. The dissociation constants $K_a^l$ are empirically corrected to the simulated temperature and ionic strength.

Using eqn. (2), one can compute the average number of protons bound to metabolite L:

$$\bar{N}_H^L = \frac{\sum_{p=1}^{N_P} p \cdot [\mathrm{H}_p\mathrm{L}]}{[\mathrm{L}_{\mathrm{tot}}]}$$

$$= \frac{\sum_{p=1}^{N_P} \frac{p \cdot [\mathrm{H}^+]^p}{\prod_{l=1}^p K_a^l}}{\mathrm{P_L}}, \qquad (3)$$

where $\mathrm{P_L}$ is the so-called binding polynomial (Alberty, 2003):

$$\mathrm{P_L} = 1 + \sum_{p=1}^{N_P} \frac{[\mathrm{H}^+]^p}{\prod_{l=1}^p K_a^l} + \sum_{q=1}^{N_q} \frac{[M^q]}{K_d^q}. \qquad (4)$$

Thus for an arbitrary biochemical reaction $k$ one obtains the following proton generation stoichiometry:

$$\Delta_r N_H^k = \sum_{\mathrm{reactants}} \overline{N}_H^{\mathrm{reactants}} - \sum_{\mathrm{products}} \overline{N}_H^{\mathrm{products}} + n^k, \qquad (5)$$

with $n^k$ denoting the number of protons generated by the associated reference reaction. In the proposed model, all reference reactions are defined in terms of the most deprotonated species in the pH range of 5.5–8.5 (cf. Table S1 in the Supplementary Material) and are balanced with respect to mass and charge (Vinnakota et al., 2006). The proton generation flux of a biochemical reaction is the product of the flux through the (reversible) biochemical

reaction $J^k$ and the proton generation stoichiometry $\Delta_r N_H^k$:

$$\mathrm{Protonflux} = \sum_{k=1}^{N_r} \Delta_r N_H^k J^k, \qquad (6)$$

where $N_r$ denotes the total number of reactions in the network. Note that changes in proton concentration and thus pH ($\mathrm{pH} = -\log_{10}([\mathrm{H}^+])$) have, in turn, an effect on the reaction fluxes $J^k$ as both the maximal velocity of a biochemical reaction, $V_{\max}^k(\mathrm{pH})$, and the apparent equilibrium constant, $K_{\mathrm{app}}^k(\mathrm{pH})$, are a function of pH as described in the following sections.

*Kinetics.* The enzyme-catalysed reaction fluxes $J^k$ are (unless specified differently) defined as reversible reactions based on Michaelis–Menten kinetics:

$$J^k = J_f^k \left( V_{\max,f}^k (\mathrm{pH}), \boldsymbol{a}, \boldsymbol{b} \right) - J_r^k \left( V_{\max,r}^k (\mathrm{pH}), \boldsymbol{a}, \boldsymbol{b} \right). (7)$$

Therein, $V_{\max,f}^k$ and $V_{\max,r}^k$ denote the maximal reaction velocity in forward and reverse directions, respectively. For enzyme reactions with a known pH optimum, $V_{\max,f}$ is described using an empirical pH correction (Vinnakota et al., 2006). Furthermore, $\boldsymbol{a}$ is a vector of substrate and product concentrations, and $\boldsymbol{b}$ contains the kinetic (Michaelis) constants for the substrates (S) and products (P).

*Thermodynamics.* The reversible reaction kinetics in the model are thermodynamically constrained through the

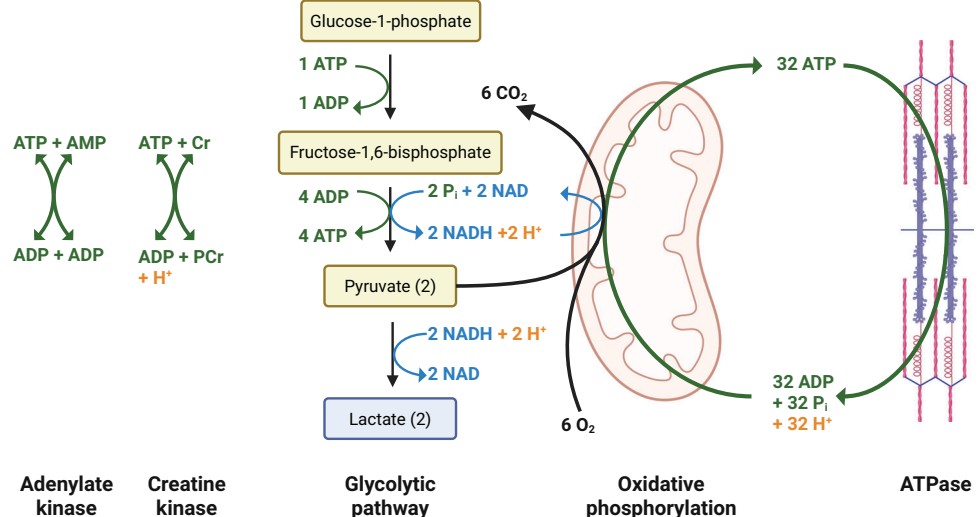

**Figure 1. Energy, redox and proton balance in FOG muscle fibres**
Schematic drawing of the biochemical network involved in the dynamic balance of ATP (green), redox state (blue) and pH dynamics (orange) in skeletal muscle fibres. This figure was created in BioRender.com.

Haldane relation, which links the apparent equilibrium constant $K_{app}^k$ to the kinetic parameters:

$$V_{\max,r}^k = \frac{V_{\max,f}^k \prod K_m^P}{K_{app}^k \prod K_m^S}.$$ (8)

Therein, $K_m^P$ and $K_m^P$ denote the Michaelis constants of substrates and products, respectively. The apparent equilibrium constant $K_{app}^k$ of reaction $k$ is defined in terms of the equilibrium constant of the associated reference reaction, $K_{ref}^k$, using the binding polynomials of the substrates ($P_S$) and products ($P_P$):

$$K_{app}^k = K_{ref}^k \cdot \frac{\prod P_P}{\prod P_S} \cdot [H^+]^{n^k}.$$ (9)

Furthermore, the equilibrium constant of each reference reaction in the model is calculated using the standard free energy of the reference reaction $\Delta_r G_k^0$, defined as the difference between the free energies of the formation of the products and reactants:

$$K_{ref}^k = e^{-\Delta_r G_k^0 / RT}.$$ (10)

Therein $R = 8.315 \text{ JK}^{-1}\text{mol}^{-1}$ is the universal gas constant and $T$ is the absolute temperature in Kelvin. The transformed Gibbs energy of each biochemical reaction can be calculated as

$$\Delta_r G_k' = \Delta_r G_k'^0 + RT \ln Q',$$ (11)

where $Q'$ is the apparent reaction quotient, that is the reaction quotient of the biochemical reaction defined in terms of the sums of species. Furthermore, the standard transformed Gibbs energy, $\Delta_r G_k'^0$, can be derived from the apparent equilibrium constant (see eqn. (9)):

$$\Delta_r G_k'^0 = -RT \ln\left(K_{app}^k\right).$$ (12)

**Muscle fibre model.** This section provides a brief description of the metabolic model describing an FOG muscle fibre. A detailed description of each reaction is provided in Section S1 of the Supplementary Material.

*ATPase activity.* We use a lumped total ATP consumption flux (e.g. cross-bridge cycling and SERCA pumps) that is modelled as the sum of a basal and an activity-dependent rate. The basal rate is fixed at 0.48 mM/min as observed in resting skeletal muscle (Blei et al., 1993). The activation rate is modelled as a step function with variable amplitude and time intervals.

*ATP buffers.* CK catalyses the reaction PCr + ADP $\rightleftarrows$ Cr + ATP, which serves as an instantaneous ATP buffer in response to changes in the ATPase activity. For the CK reaction, the free energy of formation of the reference species $HCr^0$ and $HPCr^{2-}$ was not available. Thus, instead of calculating the reference equilibrium constant

as described in eqn. (10), the apparent equilibrium was computed according to eqn. (9) using a reported literature value of $K_{ref}^{CK} = 3.77 \times 10^8 \text{M}^{-1}$ (Teague Jr & Dobson, 1992). Note that the reported $K_{ref}^{CK}$ was determined at 38°C and an ionic strength of 0.25M. These conditions deviate slightly from the proposed model with a temperature of 37°C and an ionic strength of $I = 0.1$M. Furthermore, the model captures ATP buffering through AK.

*Glycolysis.* During rest-to-work transitions, glycolytic ATP synthesis rapidly increases, and glycolysis is particularly relevant during high-intensity exercise. In this work, we describe the biochemical reaction network responsible for the turnover of glucose-1-phosphate (G1P) to pyruvate (PYR) and lactate (LAC) through various glycolytic intermediates (see Table 1). Importantly, phosphofructokinase (PFK) is a key regulatory enzyme that facilitates the rapid control of the glycolytic flux. We used a pseudorandom-order kinetic PFK model with statistical inhibition developed by Waser et al. (1983) and later adopted by Connett (1987, 1989). The PFK model accounts for activation resulting from an alkaline shift in cytoplasmic pH, as well as ATP-mediated inhibition and competitive deinhibition by the metabolites ADP and AMP. The binding constants for ADP and AMP ($K_{ADP}^{PFK}$ and $K_{AMP}^{PFK}$, see Table S2 of the Supplementary Material) were adjusted heuristically to achieve a glycolytic flux near zero in the resting state while simultaneously facilitating a rapid glycolytic flux regulation when the muscle is activated. This behaviour closely mimics experimental observations in FT muscles (Crowther et al., 2002, 2002; Dawson et al., 1980; Quistorff et al., 1993).

Note that we neglected glycogen phosphorylase (both isoforms A and B), converting glycogen into G1P, as there exists no kinetic model adequately describing the dynamics of this reaction. Instead, the G1P concentration was clamped ($[G1P] = 0.07$mM), acting as a substrate pool for phosphoglucomutase (PGLM). Theoretically, this would yield an increasing total phosphate pool (TPP). Thus, we modified the time derivative of inorganic phosphate to include the PGLM flux in a way that Pi is removed from the myoplasm at the rate of G1P turnover via PGLM (see Section S1.2 of the supplementary material).

*Mitochondrial ATP synthesis.* OxPhos in the mitochondria is the most efficient and sustainable pathway for ATP synthesis. Although mitochondria can use different fuels (e.g. fatty acids), given that we are modelling FOG fibres, we only considered oxidative carbohydrate metabolism (Crow & Kushmerick, 1982). A top–down approach was used, where all biochemical reactions associated with mitochondria, namely pyruvate dehydrogenase, TCA cycle (tricarboxylic acid cycle), electron transport chain and ATP synthase, are lumped together in one reaction (see Supplementary Material,

**Table 1. Maximal enzyme activities used in the model.**

| Enzyme | Abbreviation | $V_{max}$ (M/min) |
|---|---|---|
| Phosphoglucomutase | PGLM | 2.7[b] |
| Phosphoglucoisomerase | PGI | 1.2[a,*] |
| Phosphoglucokinase | PFK | 0.13[b] |
| Aldolase | ALD | 0.2[a] |
| Priosephosphate isomerase | TPI | 24.0[a] |
| Glycol-3-phosphate dehydrogenase | G3PDH | 1.3[b,*] |
| Glyceraldehyde-3-phosphate dehydrogenase | GAPDH | 3.3[a] |
| Phosphoglycerate kinase | PGK | 2.0[a,*] |
| Phosphoglycerate mutase | PGM | 1.6[a] |
| Enolase | ENO | 0.3[a] |
| Pyruvate kinase | PYK | 0.2[b] |
| Lactate dehydrogenase | LDH | 4.0[a] |
| Adenylate kinase | AK | 1.2[a] |
| Creatine kinase | CK | 1.0[a,*] |

[a] Reported in Eagle and Scopes (1981); [b] heuristically adjusted; * Given as reverse rate.

Section S1.1.5). This yields the following reference reaction:

$$PYR^- + NADH^{2-} + 16ADP^{3-} + 16HPO_4^{2-} + 18H^+$$
$$+3O_2 \rightarrow 16ATP^{4-} + 3CO_2 + 19H_2O$$
$$+NAD^-. \tag{13}$$

Equation (13) considers key intermediates and substrates that couple the glycolytic and aerobic pathways. Note that with limited availability of myoplasmic NADH, the capacity for ATP production by the OxPhos model is reduced, which is described in Section S1.1.5 of the Supplementary Material.

The kinetics of the OxPhos model is based on a sigmoid Hill function with feedback regulation of the substrates ADP (Jeneson et al., 1996), inorganic phosphate (Pi) and pyruvate:

$$J_{OxPhos} = V_{max}^{OxPhos} \frac{\left(\frac{[ADP]}{K_{ADP}^{OxPhos}}\right)^{n_H^{OxPhos}}}{1 + \left(\frac{[ADP]}{K_{ADP}^{OxPhos}}\right)^{n_H^{OxPhos}}} \cdot \frac{1}{1 + \frac{K_{Pi}^{OxPhos}}{[Pi]}} \cdot$$
$$\frac{1}{1 + \frac{K_{PYR}^{OxPhos}}{[PYR]}}, \tag{14}$$

where the constants $K_{ADP}^{OxPhos}$, $n_H$, $K_{Pi}^{OxPhos}$ and $K_{PYR}^{OxPhos}$ determine the sensitivity to changes of the substrate concentrations. In detail, $K_{Pi}^{OxPhos} = 1mM$ is the half-saturation concentration of Pi, whereby reported $K_m$ values from *in vitro* observations in purified mammalian heart and liver mitochondria range from 0.25 to 1 mM (Bygrave & Lehninger, 1967; Chance & Connelly, 1957). Furthermore, $K_{PYR}^{OxPhos}$ was set to 0.15 mM, a value reported to be the $K_m$ for pyruvate translocation into

rat liver mitochondria (Halestrap, 1975). This value is within the range of other reported $K_m$ values for pyruvate (Hansford, 1972; Nicklas et al., 1971; Smith & Russell, 1967; Williamson, 1964). Finally, the parameters $K_{ADP}^{OxPhos}$ as well as the maximal reaction velocity $V_{max}^{OxPhos}$ and the Hill coefficient $n_H^{OxPhos}$ were optimised using dynamic phosphorus magnetic resonance spectroscopy data as described in Section 2.2.2.

*Generic proton buffer.* In addition to the dynamic proton buffer capacity of the metabolites, the model considers buffering via, for example. proteins by assuming a constant intrinsic buffer size $[X]_T$ with a dissociation constant of $K_a = 1 \times 10^{-7}M$ for a working range around pH = 7. The buffer concentration $[X]_T$ is the sum of the acid $[HX]$ and its conjugate base $[X^-]$. The dissociation constant $K_a$ is defined as $K_a = [H^+][X^-]/[HX]$. Moreover, the buffer capacity $\beta$ is defined as the concentration of acid ($C_a$) or base ($C_b$) that needs to be added to a buffer solution to change its pH by one unit (in slykes, i.e. mmol/L/pH unit). In a physiological pH range, the buffer concentration of a weak monoprotonic acid can be calculated as follows (Urbansky & Schock, 2000):

$$\beta = \frac{dC_b}{d(pH)} \approx \ln 10 \left( [X]_T \cdot \frac{K_a[H^+]}{(K_a + [H^+])^2} \right). \tag{15}$$

*Metabolite transport.* Energy metabolism in skeletal muscles requires the exchange of products and substrates with the environment. To simulate an open system, the model is divided into three compartments, representing the muscle fibre, the extracellular fluid and the capillaries. The compartments are coupled through metabolite-specific transport systems. To couple the muscle fibre and the extracellular environment, we consider passive and facilitated transmembrane trans-

port of lactate, $CO_2$ and bicarbonate. Lactate-proton symport is achieved by the monocarboxylate transporter MCT, where we only considered isoform MCT4 (i.e. the predominant isoform in cells with a high glycolytic rate (Halestrap & Price, 1999)). Note that transport via MCT is bidirectional, allowing the utilisation of extracellular lactate generated by other cells as a fuel for the aerobic pathway. This is known as the intercellular lactate shuttle (Brooks, 1998). In this work, extracellular lactate levels were clamped to 1mM. The transport of $CO_2$ over the muscle fibre membrane occurs by passive diffusion. Furthermore, $CO_2$ and bicarbonate can be transported from the extracellular space into the capillary compartment. The kinetic equations and parameters of all transport reactions are provided in the Supplementary Material, Section S1.

**Parameters and initial conditions.** The model consists of 85 adjustable parameters (excluding the 13 apparent equilibrium constants, which are dynamically computed as described in Section 2.1.1) and 41 state variables. Most parameters could be fixed based on existing data from the literature. Only five parameters were identified by solving an optimisation problem (Section 2.2.2).

*Enzyme activities.* Maximum enzyme activities of the glycolytic reaction network, as well as CK and AK, measured at 37°C in rabbit skeletal muscle are reported in Eagle and Scopes (1981). Reported values were assumed as values per fiber volume. Four maximum enzyme activity values (PGLM, PFK, G3PDH and PYK) needed to be adjusted heuristically to handle a large range of ATP turnover rates without excessive accumulation of glycolytic intermediates under normal physiological conditions. Table 1 summarises the maximal activities for each enzyme-catalysed reaction.

*Baseline concentrations.* To obtain steady-state baseline concentrations that are consistent with experimental data from fast-twitch muscles, we set the initial concentrations of the metabolites in the model to appropriate values, primarily focusing on intracellular ATP, ADP, PCr, Pi and pH as well as intracellular lactate, as the baseline concentrations of other metabolites such as the glycolytic intermediates cannot be measured in an intact physiological system. The system is constrained by mass conservation, where the total creatine pool sums to 45.2 mM and the sum of total adenine nucleotides (TANs) is 10.32 mM. The size of the TPP is defined in terms of transferable phosphates and was fitted to experimental data as described in Section 2.2.2. Using the guessed initial concentrations, we conducted simulations at the basal ATP demand until the model reached a steady state as part of the model identification (see Section 2.2.2). The resulting steady-state baseline concentrations of the model (given the optimised parametrisation) can be found in Supplementary Material, Table S4.

*Compartments and volume fractions.* The volume fractions ($v_{my}^r$, $v_{ex}^r$, $v_{cap}^r$) and the water content ($v_{my}^w$, $v_{ex}^w$, $v_{cap}^w$) of the model compartments myoplasm, extracellular space and capillaries are needed as a scaling factor in the differential equations of each metabolite (see Supplementary Material, Section S1.2). Their values were derived using data from frog muscle (Desmedt, 1953). In short, the total water content of 1 g wet-weight frog muscle (sartorius) is 78.5 %, whereby 12.5 % can be attributed to the extracellular space and 66 % to the muscle fibre. Furthermore, we used the simplified assumption that the extracellular and capillary compartments consist of 100 % water, such that $v_{ex}^w = 1$ and $v_{cap}^w = 1$. For the myoplasm compartment, this leads to a volume fraction of $v_{my}^r = 0.875$ and water content $v_{my}^w = 0.754$. The volume fractions of the interstitial and cytoplasmic compartments are assumed to be $v_{ex}^r = 0.085$ and $v_{cap}^r = 0.04$, respectively.

## Simulations

**Computer model.** The model described in Section 2.1 yields a differential–algebraic system of equations (DAE). A numerical model was implemented using MATLAB 9.12 (Mathworks Inc., MA, USA). To obtain the right-hand side of the system of differential equations describing the biochemical reaction network, we used the MATLAB-based biochemical simulation environment (BISEN, Medical College of Wisconsin, WI, USA) (Vanlier et al., 2009). The differential equations were solved numerically using MATLAB's built-in 'ode15s' solver optimised to handle stiff initial value problems (absolute and relative tolerance: $10^{-9}$, maximal step size: 3 s). The metabolite concentrations were required to be non-negative.

**Model identification.** A set of five parameters (i.e. the maximum OxPhos rate, the Hill coefficient of the OxPhos model, the sensitivity of OxPhos to the ADP concentration, the generic buffer size and the TPP) and the ATPase activity corresponding to a specific experiment were identified by fitting the model to experimental data published by Kushmerick and Meyer (1985). In the following $q$ denotes the vector containing the model parameters and initial conditions to be estimated by solving an optimisation problem.

*Identification data.* The data used for the model identification were collected from the gastrocnemius–plantaris muscle of anaesthetised rats via $^{31}P$ magnetic resonance spectroscopy ($^{31}P$-MRS). Kushmerick & Meyer (1985) acquired dynamic $^{31}P$-MRS data before, during and after muscle activation. The muscle was activated by electrically stimulating the

sciatic nerve at frequencies of 2, 4 and 10 Hz. For each stimulation frequency, the time course of [PCr], [Pi] and intracellular pH was derived from the relative peak areas of the collected spectra.

*Optimisation problem.* To quantify the difference between model predictions and experimental data, a fitness function $L(\boldsymbol{q})$ was defined as the weighted mean absolute error (MAE) between the (regularised) experimental data $\boldsymbol{x}(t_i)$ (with $t_i$ denoting the temporal sampling points and $i = 1, \cdots, n_i$) and corresponding simulated observations $\bar{\boldsymbol{x}}(t_i, \boldsymbol{q})$. The regularisation was introduced as $^{31}$P-MRS quantifies relative changes in metabolite concentration. Thus, we compared the relative change of the simulated and measured concentrations. Approximately 1 mM of the measured Pi is considered to belong to the extracellular environment (Kushmerick & Meyer, 1985). This was taken into account by adding 1 mM Pi to the simulated time course of intracellular Pi before calculating the error between the model simulation and experimental data. Furthermore, as the simulated baseline pH depends on the selected parameters, we compared the absolute change in pH values with $\Delta\text{pH}(\boldsymbol{q})$, denoting the difference in baseline pH between the experimental data and the simulation. This yields

$$L(\boldsymbol{q}) = \frac{1}{9n_i} \sum_{m=1}^{3} \sum_{j=1}^{3} \sum_{i=1}^{n_i} \frac{\left| \bar{x}_m^j(t_i, \boldsymbol{q}) - x_m^j(t_i) \right|}{\text{range}\left( x_m^j \right)} \text{ , with}$$
(16a)

$$\boldsymbol{x}^j(t_i)\} = \left[ \frac{[\text{PCr}]^j(t_i)}{[\text{PCr}]^j(t_{\text{ref}})}, \frac{[\text{Pi}]^j(t_i)}{[\text{Pi}]^j(t_{\text{ref}})}, \text{pH}^j(t_i) \right]^T, \quad (16\text{b})$$

$$\bar{\boldsymbol{x}}^j(t_i, \boldsymbol{q}) = \left[ \frac{\overline{[\text{PCr}]}^j(t_i, \boldsymbol{q})}{\overline{[\text{PCr}]}^j(t_{\text{ref}}, \boldsymbol{q})}, \frac{\overline{[\text{Pi}]}^j(t_i, \boldsymbol{q}) + 1\text{mM}}{\overline{[\text{Pi}]}^j(t_{\text{ref}}, \boldsymbol{q}) + 1\text{mM}}, \right.$$
$$\left. \overline{\text{pH}}^j(t_i, \boldsymbol{q}) + \Delta\text{pH}(\boldsymbol{q}) \right]^T, \quad (16\text{c})$$

$$\boldsymbol{q} = \left[ V_{\text{max}}^{\text{OxPhos}}, n_{\text{H}}^{\text{OxPhos}}, K_{\text{ADP}}^{\text{OxPhos}}, [\text{X}_{\text{T}}], \text{TPP}, \right.$$
$$\left. V_{\text{act}}^{\text{ATPase}}(\text{WR}_j) \right]^T, \quad (16\text{d})$$

where the indices $m$ refer to the measured variable ($m = 1$: PCr, $m = 2$: Pi, $m = 3$: pH) and $j$ denotes the stimulation frequency ($j = 1$: 2 Hz, $j = 2$: 4 Hz, $j = 3$: 10 Hz). Furthermore, the reciprocal of the range function (i.e. the maximum minus the minimum value) of the individual time series data was used as weights.

Equation (16a) was minimised using a genetic algorithm provided by MATLAB's global optimisation toolbox and the function 'ga' (default settings). Each population contains 200 parameter sets, with the output

of the optimisation problem being the parameter set with the minimum loss in the last generation. The genetic optimisation iteratively updates the population of parameters considering (quasi-)random mutations. Hence, the obtained solution depends on the seed of the utilised random number generator as well as the initial population. To test the uniqueness of the estimated parameters, we ran the optimisation with 20 different first generations. Finally, we computed the coefficient of variation (CoV) for each parameter as well as the linear correlation between all sets of parameters.

To fit the model to the experimental data, one simulation was performed per stimulation frequency following a protocol of 1.8 min of muscle activation with an ATPase rate above basal level and a subsequent 3-min-long recovery phase. An initialisation phase at the basal ATPase rate was implemented to ensure that the model has reached a steady-state baseline with each parameter set $\boldsymbol{q}$ before the simulation protocol begins. The ATP turnover rates during muscle activation were estimated by the optimisation problem (see Table 2).

*Physiological constraints.* We restricted the solution space of the estimated parameters $\boldsymbol{q}$ by specifying lower and upper bounds corresponding to reported values from the literature (if available). The Hill coefficient $n_{\text{H}}^{\text{OxPhos}}$ of the OxPhos model (eqn. (14)) was found to be at least 2 with values reported not higher than 3 (Cieslar & Dobson, 2000; Jeneson et al., 1996; Vicini & Kushmerick, 2000). For isolated mitochondria extracted from mammalian heart muscle or liver, reported $K_{\text{m}}$ values for ADP have been typically on the order of 30 μM (Bygrave & Lehninger, 1967; Chance & Williams, 1955). Thus, the permissible parameter range of $K_{\text{ADP}}^{\text{OxPhos}}$ was chosen between 10 and 100 μM. The maximal oxidative ATP synthesis rate was computed based on the maximal increase in oxygen consumption during tetanic contraction of 3.62 μmol $O_2$ min$^{-1}$ g muscle$^{-1}$ observed in perfused rat hindlimb (Hood et al., 1986). Using ATP/$O_2$ = 5.33 (eqn. (13)) and a factor of 0.875 (Desmedt, 1953) to account for the fraction of fiber volume per unit wet weight of muscle, the $V_{\text{max}}^{\text{OxPhos}}$ value was calculated to be approximately 20/16 mM/min.

The protein buffer capacity of human muscle is estimated to be 15–45 slykes plus an additional contribution of around 3 slykes by dipeptide anserine (Kemp et al., 1993). This yields a lower and upper bound for the buffer concentration of $[\text{X}]_{\text{T,lower}} = 0.031$ M and $[\text{X}]_{\text{T,upper}} = 0.084$ M, respectively, given a basal pH value of around 7.05 (eqn. (15)).

Note that the design variables have an effect on the steady-state baseline levels of the metabolites. Yet, the ratio of PCr and Pi in the model and the experimental data needs to be similar to ensure comparability. Thus, the TPP was optimised by constraining the initial Pi concentration

**Table 2. Model parameters identified by the genetic optimisation algorithm**

| Parameter | Description | Value | CoV (%) |
|---|---|---|---|
| $V_{max}^{OxPhos}$ | Maximal reaction velocity of OxPhos | 1.25 mM/min | $\ll 1.0$ |
| $n_H$ | Hill coefficient for ADP of OxPhos | 3.0 (-) | 2.1 |
| $K_{ADP}^{OxPhos}$ | Half-saturation concentration of ADP for OxPhos | 33.388 μM | 7.4 |
| $[X]_T$ | Total concentration of intrinsic buffer (non-phosphate, non-carbonate) | 84.0 mM | 5.4 |
| TPP | Total phosphate pool | 57.7 mM | 0.9 |
| $V_{act}^{ATPase}$(2Hz)* | ATPase rate for 2 Hz stimulation | 22.2 mM/min | 5.9 |
| $V_{act}^{ATPase}$(4Hz)* | ATPase rate for 4 Hz stimulation | 41.0 mM/min | 2.5 |
| $V_{act}^{ATPase}$(10Hz)* | ATPase rate for 10 Hz stimulation | 50.9 mM/min | 1.8 |

*Model input; CoV, coefficient of variation across 20 optimization runs.

between 0 and 4 mM. This yields a permissible TPP range from 59.95 to 63.95 mM.

**Model validation.** The model was validated against independent *in vivo* $^{31}$P-MRS data of intracellular pH and PCr collected by Foley et al. (1991) during rest, muscle contraction and recovery. Gastrocnemius muscles of anaesthetised rats were stimulated at 0.75 Hz for 8 min. In contrast to the stimulation frequencies used for model identification, muscle stimulations at 0.75 Hz are below the mitochondrial ATP synthesis capacity (at approximately 0.8 Hz (Cieslar & Dobson, 2000)). The ATPase rate, that is, the model input, corresponding to a frequency of 0.75 Hz was calculated from the initial rate of PCr hydrolysis, assuming that neither glycolysis nor OxPhos has a significant contribution to ATP supply right at the onset of muscle stimulation. The ATPase rate was reported to be $11.51 \pm 1.47$ μmol g$^{-1}$min$^{-1}$ calculated from a monoexponential fit to the measured PCr changes (Foley et al., 1991). Using a factor of 0.875 (Desmedt, 1953) to account for the fraction of fiber volume per unit wet weight of muscle, the calculated ATPase rate is $13.15 \pm 1.68$ mmolL$^{-1}$min$^{-1}$ and thus below the model's maximal capacity for oxidative ATP synthesis of 20 mmolL$^{-1}$min$^{-1}$.

We also estimated the ATPase rate during exercise by fitting the model output directly to the measured PCr dynamics (Foley et al., 1991) using a genetic algorithm and the same method as for the parameter identification described in the section above. Note that in this case, there is only one stimulation frequency ($j = 1$: 0.75 Hz) and one measured variable ($m = 1$: PCr) considered in $L(\boldsymbol{q})$ and consequently $\boldsymbol{x}^j(t_i)$ and $\boldsymbol{x}^j(t_i, \boldsymbol{q})$. Further, the ATPase rate was the only design variable in $\boldsymbol{q}$ was determined to be 12.06 mmolL$^{-1}$min$^{-1}$.

**Multiparametric sensitivity analysis.** A local multiparameter sensitivity analysis (MPSA) (Chang & Delleur, 1992; Choi et al., 1998; Hornberger & Spear, 1981; Zi et al., 2005) was performed to evaluate the sensitivity of the model to (simultaneous) variations of kinetic parameters and the total phosphate, creatine, adenosine and redox pool sizes as well as the intrinsic proton buffer size. In short, the MPSA is based on a Monte Carlo method in which the model is run repeatedly using different parameter sets drawn from a uniform probability distribution for each tested parameter. All parameters were varied by $\pm 1\%$. The sensitivity values $D_{m,n} \in [0, 1]$ are computed using the Kolmogorov–Smirnov statistic and quantify the influence of variations of a parameter $n$ on a state variable $m$, with sensitivity values close to 1 indicating high sensitivities. A detailed step-by-step description of the performed MPSA can be found in the Supplementary Material, Section S2.

**Model predictions**

*Experiment 1: Intracellular redox potential.* Currently, no *in vivo* modality can directly measure internal state variables, such as the concentration of glycolytic intermediates or the cytoplasmic redox potential. We use the proposed computational model to predict the internal dynamics of the given biochemical network. For this purpose, we simulated 10 min of muscle activation with variable contraction intensities, corresponding to ATPase rates ranging from 0.5 to 30 mM/min in steps of 0.5 mM/min, followed by a recovery phase with an ATPase rate at the basal level.

*Experiment 2: LDH-KO.* Lactate dehydrogenase (LDH) catalyses the reversible reduction of pyruvate to lactate while simultaneously oxidising NADH. To study the role of LDH in ATP homeostasis and redox balance, we built an LDH knockout model (i.e. $V_{max}^{LDH} = 0$ mM/min). We simulated a 20-min-long exercise protocol at different ATPase rates between 0.5 and 20 mM/min, that is, the maximal capacity of oxidative ATP production, with a step size of 0.25 mM/min. As the knockout of LDH influences the basal state of the muscle fibre, the knockout model's resting state was determined by running the model without applying a stimulus before the simulation

protocol. During the initialisation phase, the pH value was clamped since the results from the MPSA indicate a strong influence of pH on the model baseline (see Section 3.1.3). For comparison, we applied the same exercise protocols to the control model with normal LDH activity (i.e. $V_{\max}^{\mathrm{LDH}} = 4000\,\mathrm{mM/min}$).

*Experiment 3: LDH-KO under ischaemia.* Furthermore, we investigated the impact of perturbations in myoplasmic redox balance on ATP metabolism following the knockout of LDH under ischemic conditions. Ischemia was simulated by setting $V_{\max}^{\mathrm{OxPhos}}$ to 0 mM/min at the onset of muscle activation. In addition, the removal of lactate and carbon dioxide from the extracellular space was prevented by unclamping extracellular lactate and setting transport mechanisms from the extracellular to the capillary compartment to zero. We compared three different model configurations, namely the (healthy) control model, the LDH KO model and an LDH KO model with clamped $NAD^+$ and NADH concentrations, when prescribing an ATPase rate of 20 mM/min with a duration of 1 min.

## Results

### Model development

**Parameter identification.** A few model parameters that could not be determined through literature values were fitted to experimental data (Kushmerick & Meyer, 1985) using a genetic optimisation algorithm (see Section 2.2.2). The optimised parameter set with the smallest loss function value, obtained from the 20 optimisation runs, was used for the parametrisation of the model. The parameter values are given in Table 2 together with the CoV computed across the 20 optimisation runs. Notably, the CoV is not bigger than 7.4 % for all estimated parameters, demonstrating the robustness of the parameter identification method.

Figure 2 shows the computed myoplasmic PCr, Pi and pH dynamics in response to step changes in ATP turnover rate for three different ATPase rates *versus* experimental data collected *in vivo* from electrically stimulated rat hindlimb FT muscle (Kushmerick & Meyer, 1985) after fitting of five adjustable parameters and the stimulation-specific ATPase rates (Table 2). Overall, the model captures the measured dynamics of myoplasmic PCr and Pi concentrations and pH levels. Due to ATP hydrolysis and immediate resynthesis through the CK reaction, the Pi and PCr dynamics show a strong negative correlation. Notably, both the experimental data and the model show second-order underdamped behaviour for PCr and Pi concentrations in the transition from resting to active state, manifested as a transient oscillation approximately 30 s after onset of contractions (Fig. 2*A–F*) rather than

a first-order mono-exponential behaviour. While the *in silico* predictions match the *in vivo* data quantitatively at 2 Hz (MAE = 1.5 mM), the model underestimates the first Pi peak at 4 Hz (MAE = 3.1 mM) and both Pi peaks at 10Hz (MAE = 4.0 mM). Furthermore, the simulated PCr dynamics at 2 and 4 Hz appear in agreement with the identification data (Fig. 2*A* and *B*), but not the 10 Hz contractions, where the experimental data show PCr recovery after 1 min of exercise. Considering myoplasmic pH dynamics, the model captures the initial net alkalinisation followed by net acidification and subsequent recovery (Fig. 2*G–I*). The absolute change in myoplasmic pH measured in rodent FT muscle in response to serial contractions, however, exceeds the model predictions for all three contraction frequencies, in particular at 2 Hz.

Finally, we computed the correlation between the estimated parameters (see Fig. A.1). The ATPase rates show a strong positive correlation with the generic proton buffer capacity $[X]_T$ ($r \geqq 0.74$; $p \ll 0.001$). This is accompanied by a strong positive correlation between the individual ATPase rates ($r \geqq 0.78$; $p \lll 0.001$). Furthermore, a strong negative correlation is detected between the TPP and $K_{\mathrm{ADP}}^{\mathrm{OxPhos}}$ ($r \leqq 0.9$; $p \lll 0.001$). Finally, we note that the values of $V_{\max}^{\mathrm{OxPhos}}$ and $n_{\mathrm{H}}^{\mathrm{OxPhos}}$ are equal to the prescribed upper bound values.

**Model verification.** To verify the metabolic muscle fibre model, *in silico* predictions were compared to independent *in vivo* [31]P-MRS data of pH and PCr dynamics collected *in vivo* from electrically stimulated FT cat biceps muscle (Foley et al., 1991) (see Fig. 3). First we ran the simulation at the reported ATP turnover rate of 13.5 ± 1.68 mM/min associated with electrical stimulation at a frequency of 0.75 Hz that the authors derived from a mono-exponential fit to the measured PCr concentration (Foley et al., 1991) (Figure 3, black line). For this setting, the model qualitatively reproduced the measured PCr and pH dynamics but underestimated the initial change in myofibrillar pH both at the onset of muscle activation and recovery (Fig. 3). Treating the ATPase rate as an adjustable parameter, an optimisation run yielded an ATPase rate of 12.06 mM/min, which is within one standard deviation of the reported ATP turnover rate (Fig. 3, blue line).

**MPSA.** To identify the parameters that have the largest influence on the model output and to test the robustness of the model given small perturbations of the parameters, we performed a local MPSA (see Section 2.2.4). Figure 4 summarises the results obtained from the MPSA. Only parameters with sensitivity values $D_{m,n} \geqq 0.15$ (with 1 indicating strong sensitivities and 0 denoting completely insensitive parameters) for at least one state variable at

one of the muscle activation levels are presented. In the resting state, all baseline concentrations excluding ATP are most sensitive to pH ($D_{m,n} \geq 0.53$). The basal ATP concentration is most sensitive to the size of the TAN pool ($D_{m,n} = 0.49$). Considering only active conditions, all state variables are most sensitive to the basal pH value ($D_{m,n} \geq 0.37$). Furthermore only modest sensitivities of the model dynamics during muscle contraction are observed for variations of the phosphate ($D_{m,n} \leq 0.23$), creatine ($D_{m,n} \leq 0.17$) and adenine pool size ($D_{m,n} \leq 0.12$), respectively.

## In silico studies

### Experiment 1: cellular redox potential

*Temporal dynamics.* The *in silico* model was used to predict the dynamics of the internal state variables at different contraction intensities (from 2.5 % to 150 % of the maximal oxidative ATP production capacity). Figure 5 exemplarily illustrates the model response for a low, medium and high intensity contraction. This corresponds to ATPase rates below (50 %), right at (100 %) and above (150 %) the maximal oxidative ATP synthesis rate (20 mM/min). While ATP content is stable (approx. 10.3 mM) during all exercise protocols (Fig. 5*B*), the PCr concentration drops by 46 %, 69 % and 92 % during the low, medium and high intensity work rates, respectively (Fig. 5*A*). Concurrently, the CK reaction remains close to equilibrium ($0.95 \leq J^{CK}_{for}/J^{CK}_{rev} \leq 1.1$, results not shown). Due to ATP hydrolysis and immediate resynthesis through

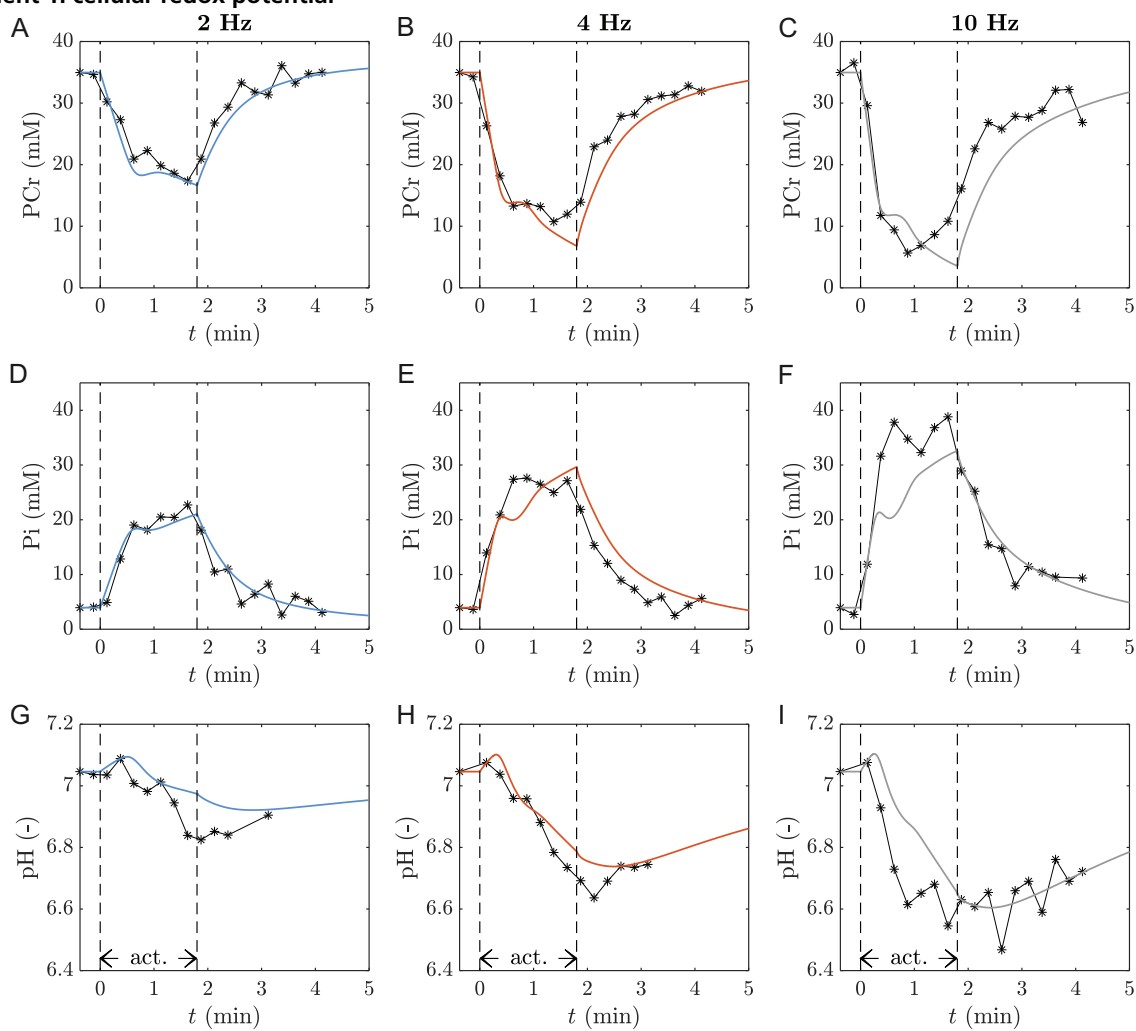

**Figure 2. Model predictions vs identification data**
Comparison between the optimized model output (solid lines) and measured data (markers) (Kushmerick and Meyer, 1985) (*A–C*) [PCr], (*D–F*) [Pi] and (*G–I*) pH dynamics depending on the stimulation frequency (Left: 2 Hz; Middle: 4 Hz; Right: 10 Hz). Note that the visualised data are normalised such that the simulated and measured data have the same baseline, see Section 2.2.2.

the CK reaction, the PCr decrease is approximately stoichiometric with the 6-fold (low intensity), 9-fold (medium intensity) and 11-fold (high intensity) increase in Pi.

After an initial alkalinisation, the pH values decrease to 7.0, 6.8, and 6.5 during low intensity, medium intensity, and high intensity exercise, respectively (Fig. 5*C*). The initial alkalinisation can be attributed to balancing of ATPase activity by PCr breakdown and the subsequent acidification to the contribution of glycolytic ATP re-synthesis to ATP balance (Fig. A.4*A*,*D* and *G*). Considering individual pathways, the main proton generation flux is caused by ATPase itself (Fig. A.4*B*,*E*, and *H*). Importantly, the effective proton generation stoichiometry of each reaction is pH-dependent (see Fig. A.5). For example, the proton stoichiometry factor of ATP hydrolysis is 0.74 at basal pH and −0.02 at pH 6.0. For glycolysis, the proton stoichiometry factor is −0.21 and 2.05 at pH 7.0 and pH 6.0, respectively (assuming there is no accumulation of glycolytic intermediates). Hence, the contribution of glycolysis to the myoplasmic proton generation flux increases over time and decreases for ATP hydrolysis.

The dynamics of the intracellular NADH concentration are presented in Fig. 5*D*. Directly after the onset of muscle contraction, a transient increase in the NADH concentration is observed. This is due to a slight imbalance between NAD reduction in the GAPDH and LDH (small reverse flux) reactions and the NADH oxidation through primarily OxPhos but also G3PDH (Fig. A.2*A*–*C*), resulting in a total net NADH generation flux in the micro molar per minute range (Fig. A.2*D*–*F*). After a few seconds of muscle activation, an increase in the metabolic flux through GAPDH and subsequently through LDH, as well

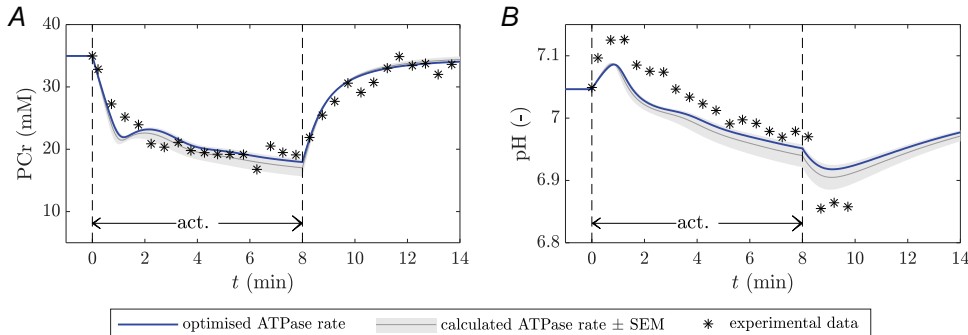

**Figure 3. Model predictions vs independent verification data**
Comparison of simulations and independent experimental data (Foley et al., 1991). (*A*) [PCr] and (*B*) pH dynamics in response to an 8-min exercise protocol (electric stimulation with frequency 0.75 Hz). Black markers: experimental data; shaded grey area: simulations with the range of ATPase rates given in Foley et al. (1991); blue lines: simulations with optimised ATPase rate. Note that the visualised PCr data are normalised such that the simulated and measured data have the same baseline.

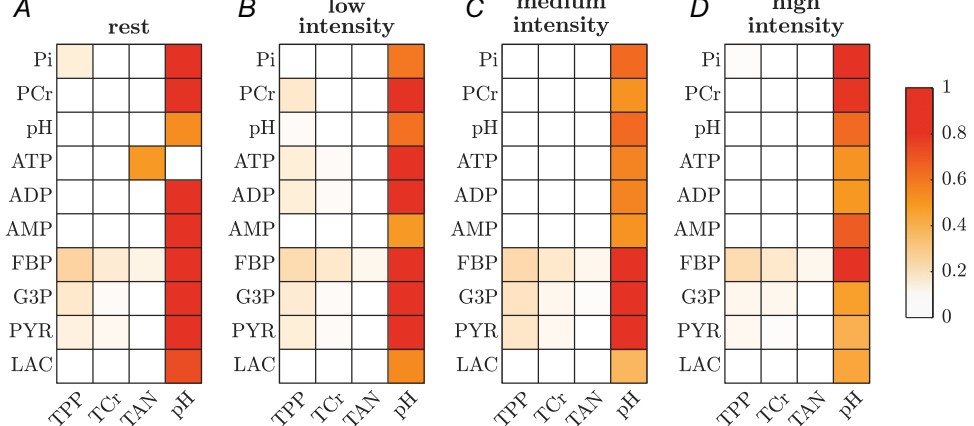

**Figure 4. Summary of the multiparameter sensitivity analysis (MPSA)**
The heat maps show the Kolmogorov–Smirnov (K–S) statistic ($D_{m,n} \in [0, 1]$) between pairs of state variables $m$ and parameters $n$, whereby higher values indicate stronger sensitivities. Columns represent the tested parameters with $D_{m,n} \geq 0.15$ with respect to at least one of the state variables at any one of the simulated work rates. Rows show selected state variables. (*A*) sensitivity of the basal state. (*B*–*D*) sensitivity during muscle activations of variable intensity.

as activation of OxPhos, is observed. This leads to NADH re-oxidation, such that the concentration drops below the basal level to approximately 50 % of the initial NADH concentration for all three work rates. The NAD reduction via GAPDH is partially buffered by G3PDH. After a transient phase (i.e. approximately the first 2 min of muscle activation), the model virtually reaches a steady state.

*Steady state.* For all contraction intensities, at the end of exercise ($t = 10$ min), the ATP re-synthesis is dominantly achieved through OxPhos (low intensity: 83 %, medium intensity: 74 %, high intensity: 59 %), see Fig. 6*A*. Nevertheless, for the low intensity and medium intensity exercise below and at the maximal oxidative ATP synthesis capacity, the glycolytic pathway contributes considerably to ATP synthesis. Moreover, even for the low intensity exercise, NAD$^+$ resynthesis at $t = 10$ min is approximately equally distributed between OxPhos and LDH (Fig. 6*C*). For medium and high intensity contractions, the LDH contribution dominates. Notably oxidative pyruvate and NADH flux during low intensity exercise (i.e. 50 % of the maximal oxidative capacity) is less than half of the maximum capacity of OxPhos for consuming NADH and pyruvate (i.e. 1.1 mM/min, whereby one makes use of the fact that the stochiometry factor of NADH and pyruvate is one, see eqn. (13)). During high intensity exercise, this value increases to approximately 80 % of the maximal flux rate of the OxPhos reaction. Finally, Fig. 6*B* and *D* show the transformed

Gibbs reaction energy of the ATPase and NAD$^+$/NADH redox (half) reaction (see Eqn. (11)), depending on the contraction intensity. As the redox half reaction is not a stand-alone reaction in the model, $K_{app}^{redox}$ was calculated manually as described in Appendix A.3. It is observed that the change in the transformed Gibbs energy of ATP hydrolysis is around three times larger than the change in the Gibbs energy of the redox couple NAD$^+$/NADH.

Myoplasmic acidification at the end of exercise ($t = 10$ min) can be attributed to energy demand being partially balanced by glycolytic ATP synthesis (Fig. 7*A*). The primary source of H$^+$, however, is ATP hydrolysis itself (Fig. 7*B*). Although net proton production by glycolysis continuously increases with ATP demand, its contribution to acidification is comparatively small, specifically for exercise intensities below the maximal capacity for oxidative ATP production. The overall proton production in the myoplasm is counteracted by the proton consumption of primarily OxPhos, as well as the monocarboxylate transporter, with increasingly higher exercise intensities. Overall proton fluxes are tightly balanced in the myoplasm as, compared to the overall induced acidification by the individual reactions, the net change in free protons is relatively small (see red area in Fig. 7*A*,*B*). Similarly while the overall proton production flux within the glycolytic pathway, mainly driven by the GAPDH or PGK reaction, exceeds the proton production flux of ATPase for high intensity workloads near 30 mM/min, other reactions in the pathway, such as the LDH reaction

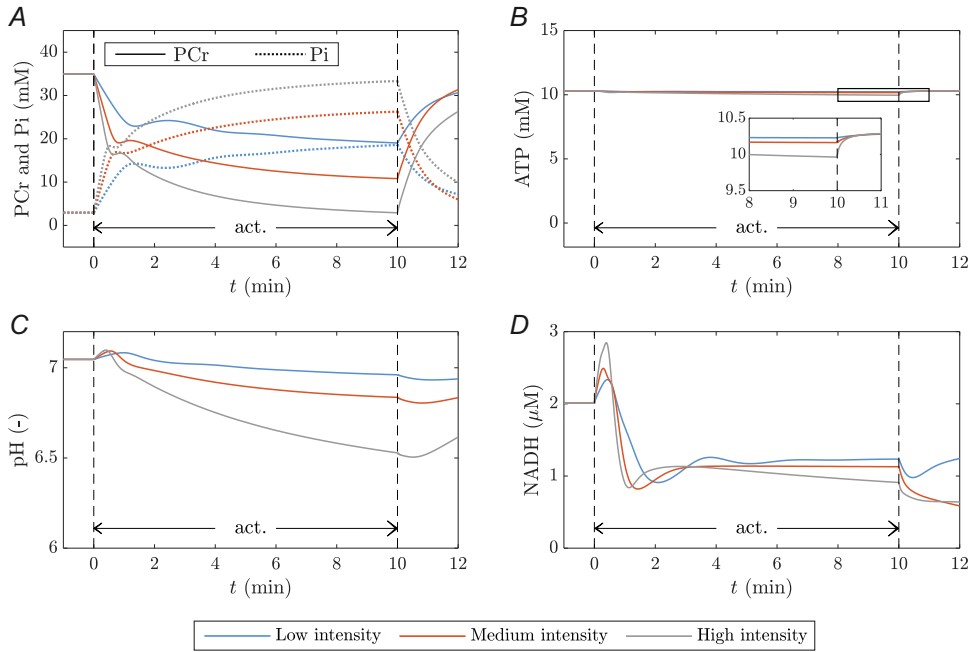

**Figure 5. ATP metabolism, pH and redox dynamics for different workloads**
Model dynamics for three different contraction intensities (Blue: ATPase rate 10 mM/min, Orange: ATPase rate 20 mM/min, Grey: ATPase rate 30 mM/min). We show the measurable variables (*A*) [PCr], (*B*) [Pi] and (*C*) pH together with the non-measurable (*D*) myoplasmic [NADH].

and PYK reaction, counteract the proton production considerably leading to a rather small net acidification by glycolysis (Fig. 7*C*).

**Experiment 2: LDH knockout**

*Temporal dynamics.* Figure 8 exemplarily compares the behaviour of the control model and the LDH-KO model given a 20-min-long exercise protocol with an ATPase rate at the maximal oxidative capacity of ATP production ($J_{ATP,max}^{OxPhos} = 20$ mM/min). Both model configurations show stable ATP concentrations during the entire exercise protocol (Fig. 8*A*). The LDH-KO simulation lacks accumulation of lactate, whereas the intracellular lactate content in the control simulation increases to approximately 8.5 mM (Fig. 8*D*). Furthermore both the control model and the LDH-KO model show a transient reduction of the cytoplasmic redox state at the onset of muscle activation (Fig. 8*B*). The LDH-KO model shows a higher maximum NADH concentration (LDH-KO: 3.8 µM; Control model: 2.5 µM) and requires more time to reach a steady-state NADH concentration during exercise. Moreover, in the control model, the steady-state NADH concentration during exercise is below the baseline concentration ($\Delta$[NADH] $\approx -0.9$ µM), and for the LDH-KO model, the steady-state NADH concentration is

above the basal level ($\Delta$[NADH] $\approx 0.6$ µM). Finally we note that the difference in the predicted pH dynamics is marginal (Fig. 8*C*), with both systems dropping to a pH value of 6.8 (MAE = 0.006).

*Steady state.* We simulated 20-min-long contractions with variable intensities (up to the maximum oxidative capacity of ATP synthesis, $J_{ATP,max}^{OxPhos} = 20$ mM/min) to predict the steady-state contributions of OxPhos and glycolysis to the overall ATP synthesis rate during exercise. The steady-state values were determined at the end of muscle activation ($t = 20$ min) and are summarised in Fig. 9*B,E*. For the control model and at the lowest considered ATPase rate (0.5 mM/min), OxPhos covers 98 % of the ATP demand. This value drops to 84 % and 75 % for ATPase rates of 10 and 20 mM/min, respectively. In the LDH-KO model, around 88 % of ATP demand is covered through OxPhos for all simulated ATPase rates.

Figure 9*C,F* shows the relative steady-state contribution of GAPDH, LDH and OxPhos to cytoplasmic redox balance during exercise. In the control system and at the lowest considered ATPase rate (0.5 mM/min), LDH runs backward utilising extracellular lactate to generate pyruvate (though the flux is close to zero with $-0.03$ mM/min). In that case, 91% of total NAD reduction can be attributed to LDH. In contrast, for ATP demands

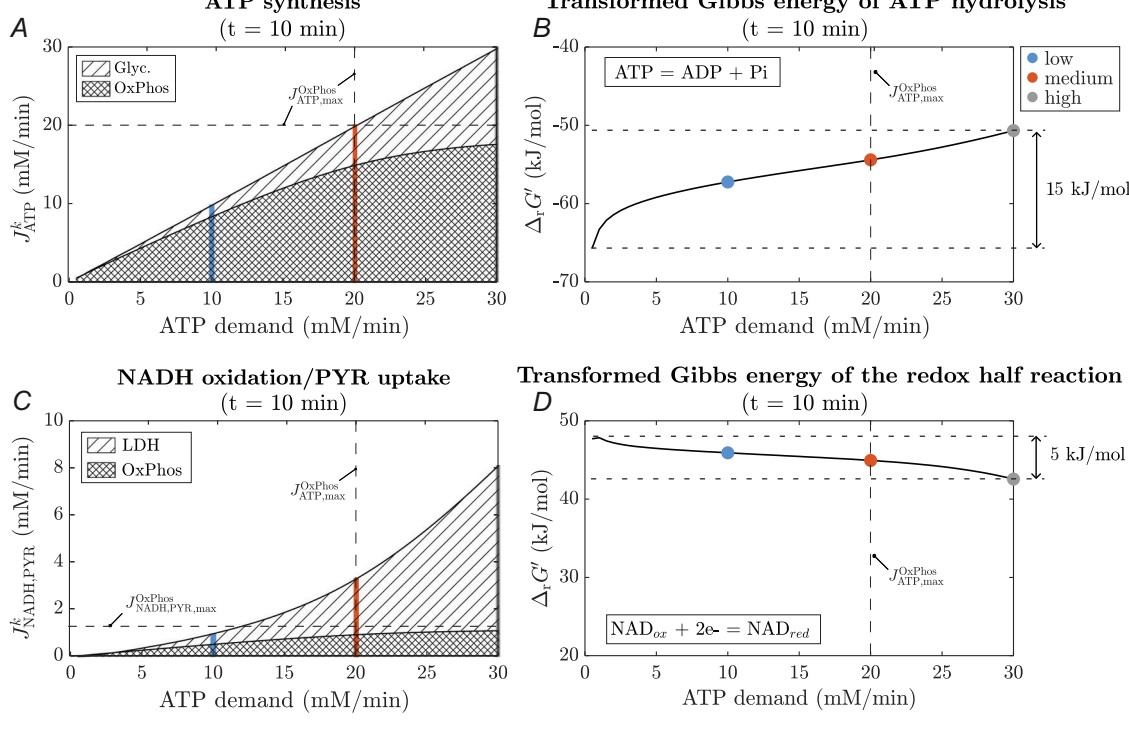

**Figure 6. Steady-state ATP and redox balance**
Contribution of the glycolytic pathway and OxPhos to (*A*) ATP synthesis and (*C*) NADH oxidation/PYR uptake as well as transformed Gibbs energy of (*B*) ATP hydrolysis and (*D*) the redox half reaction at the end of 10 min exercise at ATPase rates ranging from 0.48 mM/min (rest) to 30 mM/min. Colours indicate different ATPase rates: Blue: 10 mM/min, Orange: 20 mM/min, Grey: 30 mM/min.

above 3 mM/min, LDH converts pyruvate into lactate and, thus, together with OxPhos, oxidises NADH. The relative contribution of LDH to NADH oxidation is around 48 % (LDH flux: 0.49 mM/min) and 72 % (LDH flux: 2.4 mM/min) for ATP demands of 50 % and 100 % of the oxidative ATP production capacity, respectively. Consequently, the metabolic system utilises anaerobic glycolysis at ATPase rates considerably below the maximal oxidative ATP synthesis capacity. In the LDH-KO model, 100 % of NADH generated by the GAPDH reaction is reoxidised via OxPhos.

**Experiment 3: LDH knockout under ischemia.** To further study how myoplasmic redox balance impacts metabolite fluxes through the glycolytic pathway, we simulated 1 min ATP metabolism at an ATPase rate of 20 mM/min under ischaemic conditions, comparing the control model *vs.* the LDH-KO model. Figure 10($A$–$C$) shows the averaged fluxes through each enzyme-catalysed reaction in the glycolytic pathway during muscle activation. Comparing the simulations of the LDH-KO model (Fig. 10$B$) with the control model (Fig. 10$A$), the glycolytic fluxes are elevated, particularly in the proximal part. For example, the flux

through PFK increases by 409 %, while the reaction rate through PYK is increased by 42 %. Furthermore, the LDH-KO simulation shows an increased activity of G3PDH by 680 % compared to the control system. The model predicts a 10-fold increase in the intracellular NADH concentration (Fig. 10$E$) as well as an accumulation of glycolytic intermediates (Fig. 10$G$) for the LDH-KO system during exercise. Compared to the control model, there is no lactate accumulation, the pyruvate concentration at the end of muscle activation is 8.3 mM higher, the fructose 1,6-phosphate concentration increases by 10.5 mM and the glycerol 3-phosphate concentration is 7.3 mM higher. The altered dynamics in the LDH-KO model result in a negative net production of ATP in the glycolytic pathway, as ATP consumption in the proximal part exceeds ATP production in the distal part. That is, the model predicts a glycolytic ATP synthesis rate of −22.9 mM/min at the end of muscle activation compared to 17.0 mM/min in the control system. As a result, the system is not able to match the energy demand, leading to a 2 mM decrease in ATP (Fig. 10$D$).

To test the role of myoplasmic redox balance in maintaining sustainable glycolytic ATP production, we

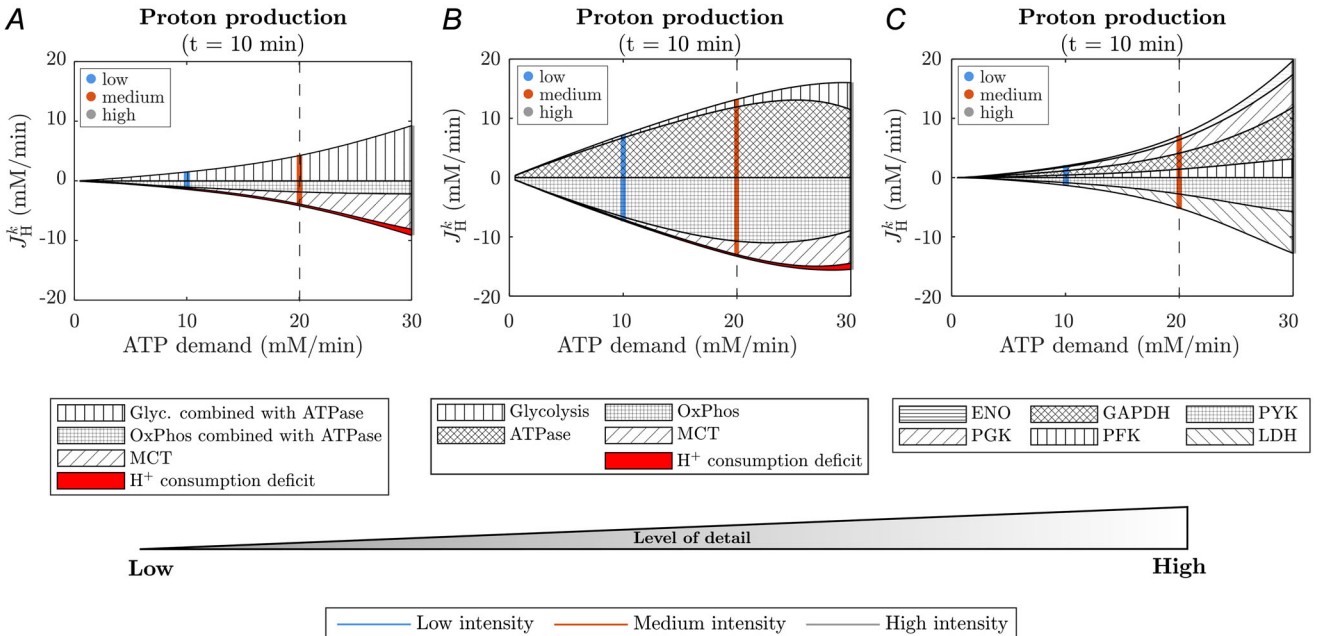

**Figure 7. Multiscale analysis of steady-state proton balance**
Contribution to myoplasmic proton production at the end of 10 min exercise at ATPase rates ranging from 0.48 mM/min (rest) to 30 mM/min. A positive flux indicates proton production; a negative flux indicates proton consumption. ATPase rates corresponding to low, medium and high intensity exercise are marked with different colours: Blue: 10 mM/min, Orange: 20 mM/min, Grey: 30 mM/min. The vertical dashed line marks the maximal capacity for oxidative ATP production. ($A$) Proton flux of lumped pathways, that is, the glycolytic pathway combined with the fraction of ATPase balanced by glycolytic ATP synthesis, and OxPhos combined with the fraction of ATPase balanced by oxidative phosphorylation. The red area indicates the overall myoplasmic proton consumption deficit that leads to the observed acidification. At $t = 10$ min, CK, AK, carbonic anhydrase and $CO_2$ diffusion have a proton flux near zero and are not shown. ($B$) Proton flux of the individual pathways. ($C$) Proton flux of selected glycolytic enzymes.

simulated an LDH-KO model with clamped NAD$^+$ and NADH concentrations (Fig. 10$C$). This model configuration (LDH-KO, clamped redox) shows highly similar fluxes to those of the control model. For example, the PFK flux is elevated by 3 % and PYK reaction rate shows no alteration. In particular, the activity of G3PDH only shows a moderate increase of 13 %. Accordingly, compared to the control model, there is no considerable accumulation of glycolytic intermediates (Fig. 10$G$). The most notable difference is the increased level of pyruvate and the absence of lactate accumulation, both of which are concomitant effects of the LDH knockout. Analogous to the control model, the model with clamped NAD$^+$ and NADH is able to maintain ATP homeostasis (Fig. 10$D$) with a positive net production of ATP in the glycolytic pathway. At the end of the 1-min-long exercise, the glycolytic ATP synthesis rate is 11.9 mM/min. The pH dynamics are similar for all three model configurations (Fig. 10$F$).

## Discussion

This study identified and validated a computational model of carbohydrate metabolism in active FOG myofibres based on dynamic *in vivo* $^{31}$P-MRS recordings of ATP metabolite and pH time courses. Our simulations show that a strict feedback-driven biochemical control scheme suffices to explain the ATP supply–demand relationship over a roughly 100-fold range of ATP demands. Our model additionally provides insight into the concomitant

dynamics of metabolic intermediates, including pyruvate and myoplasmic NADH, that have proven difficult to track experimentally and informs understanding of the role of pyruvate and lactate metabolism in skeletal energetics. For example, our analysis predicts that the contribution of LDH to the NADH redox balance is crucial in maintaining ATP homeostasis when metabolic demand exceeds the capacity for oxidative ATP synthesis. In fact, we conclude that LDH's function in maintaining relatively low cytoplasmic NADH levels is fundamentally more important than its role in counteracting cellular acidification under high ATP demand.

### ATP turnover-driven carbohydrate metabolism in the FOG myofibre phenotype

Our computer simulations of the metabolic response in a FOG myofibre to step changes in myoplasmic ATP demand first of all showed that the combined action of carbon mass flow and feedback control of OxPhos embedded in the model suffices to maintain stable ATP and energetic state over a 100-fold range of ATP turnover rates (Fig. 2). The predicted relative contributions of glycolytic and oxidative ATP synthesis flux to total ATP production over ATP turnover rates spanning from basal rate to 1.5-fold above the maximal rate of mitochondrial OxPhos (Fig. 6$A$) are in line with estimates from experimental *in vivo* studies (Jeneson et al., 1997; Kemper et al., 2001; Walter et al., 1999). Likewise, the predicted magnitude of changes in myo-

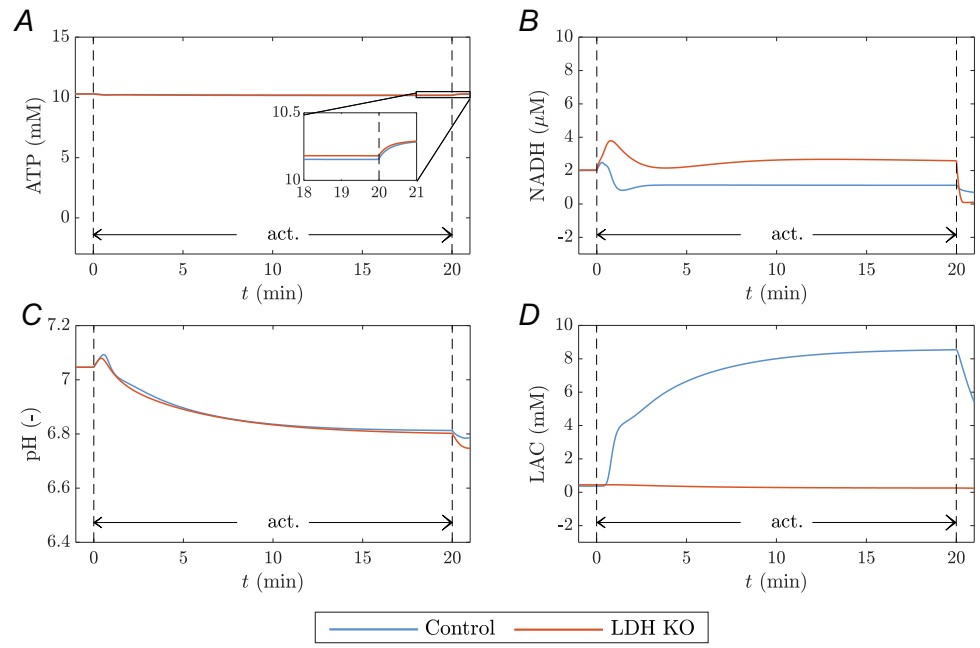

**Figure 8. Effect of LDH-KO on ATP, proton, and redox dynamics**
Comparison of the control model (blue) and LDH KO model (orange). (*A*) [ATP], (*B*) [NADH], (*C*) pH and (*D*) [LAC] dynamics during a 20-min exercise at an ATPase rate of 20 mM/min.

plasmic PCr and Pi concentrations and pH with increasing ATP turnover (Fig. 5) associated with a 15 kJ/mol drop in free energy of myoplasmic ATP hydrolysis over this operational range (Fig. 6) are similar to *in vivo* experimental observations in skeletal muscles composed predominantly of FT myofibres (Hancock et al., 2005; Habets et al., 2022; Kushmerick et al., 1992; Vandenborne et al., 1991). As such, these results support the contention of previous *in silico* studies of energy balance in skeletal muscle that feedback control of mitochondrial OxPhos suffices as a coarse regulatory mechanism to maintain ATP energy balance in this tissue (Jeneson et al., 2000; Wu et al., 2007). Earlier studies proposed that some form of feedforward control of mitochondrial ATP synthesis must also be operational *in vivo* to balance high rates of cellular ATP turnover (Korzeniewski, 2000; Vicini & Kushmerick, 2000). However, these studies only considered feedback control by ADP. It has since been shown that Pi contributes to respiratory control in FT myofibres (Jeneson et al., 2011; Schmitz et al., 2012) and was included in the present model.

Secondly, step response testing of our model identified second-order underdamped dynamic behaviour of the network in transitions between steady states, manifest, for example, in the time course of myoplasmic concentrations of PCr, Pi and NADH (see e.g. Fig. 5). This behaviour was observed at all tested ATP turnover rates, but most prominently at low ATP turnover rates. Its origin is a transient mismatch between carbon mass flow through glycolysis and mitochondrial OxPhos, respectively, resulting from ADP stimulation of both PFK and OxPhos (Section S1.1.3 of the Supplementary Material and main text Section 2.1.2). This interpretation is confirmed by the finding that increasing $V_{max}$ of OxPhos or LDH knockout aggravated the second-order behaviour by increasing pyruvate mass flow to mitochondrial metabolism (Figs. A.8 and A.6) while decreasing mass flow through lower glycolysis. Increasing $V_{max}$ of G3PDH had the opposite effect (Fig. A.8). Comparison of model simulations of PCr and Pi dynamics to experimental *in vivo* $^{31}$P-MRS recordings from FT muscle at low to moderate electrical stimulation frequencies tends to support the model prediction of second-order behaviour (Fig. 2*A,B* and *D,E*) but this interpretation is inconclusive due to sensitivity limitations inherent to dynamic *in vivo* $^{31}$P-MRS that typically preclude any sampling of P-metabolite time courses with both high time resolution and high signal-to-noise $^{31}$P-MRS (Meyerspeer et al., 2021). Alternatively, this particular model prediction may be interpreted as demonstrating that kinetic controls of PFK and other glycolytic enzymes, not captured by our model, in fact play a role in fine-tuning carbon mass flow through upper and lower glycolysis *in vivo*. Specifically, our model does not capture the complete set of known allosteric modifiers of the enzymatic activity of PFK and GAPDH (Choe et al., 2025) nor any of the proposed

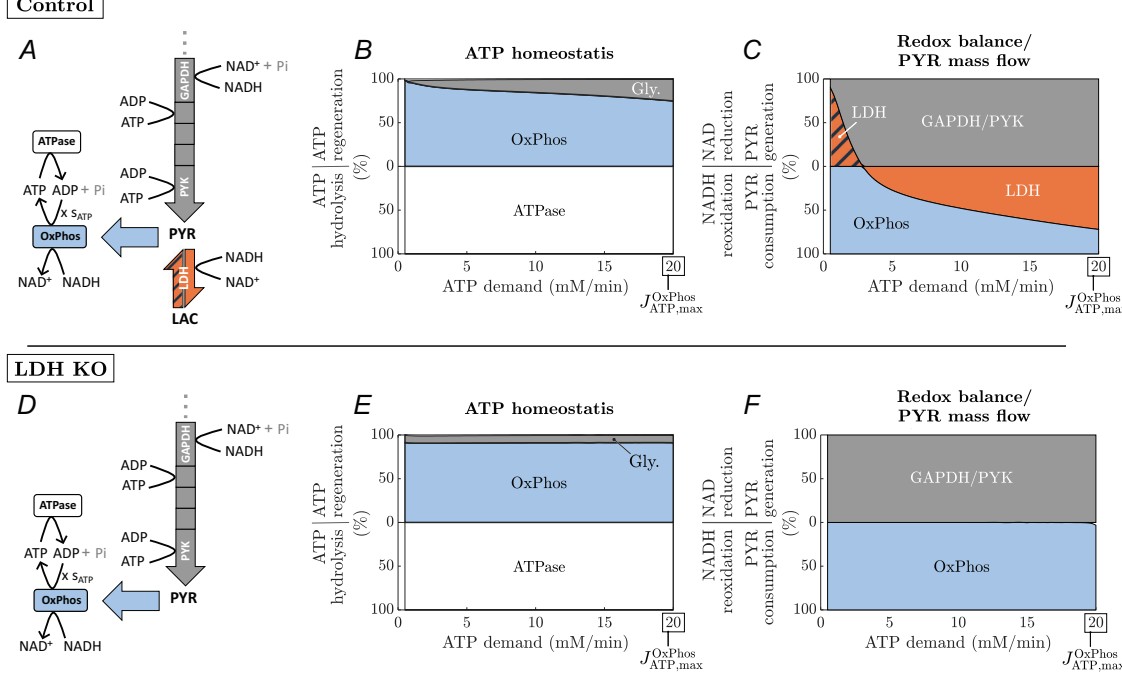

**Figure 9. Effect of LDH-KO on steady-state ATP and redox balance**
Comparison of (*A*) control model and (*D*) LDH KO model. Steady-state relative contribution to (*B, E*) ATP synthesis and (*C, F*) redox balance/pyruvate mass flow at the end of a 20-min exercise at ATP demands ranging from 0.5 mM/min to the theoretical maximal mitochondrial ATP synthesis rate of 20 mM/min.

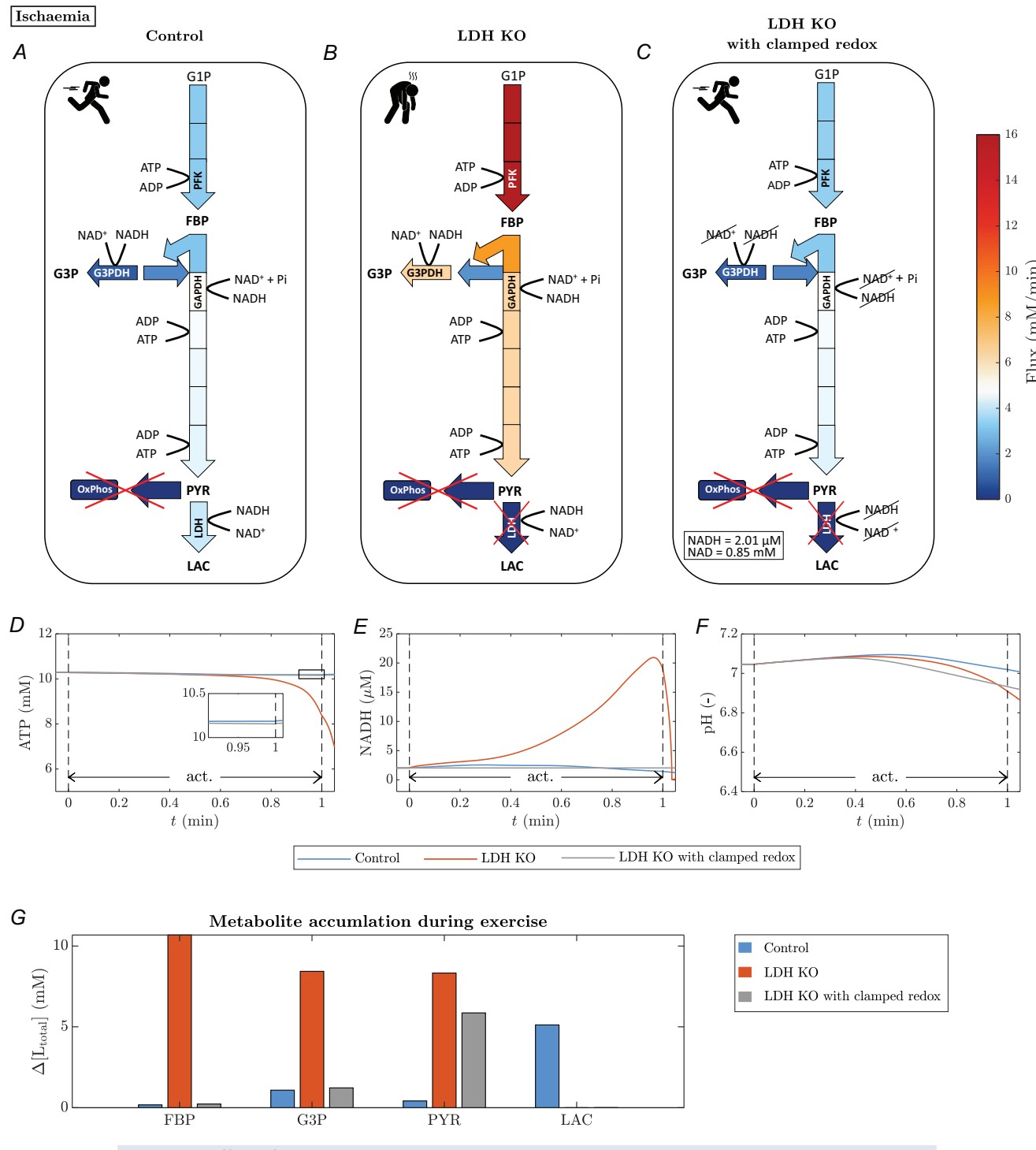

**Figure 10. Effect of LDH-KO in ischaemic conditions**
(*A–C*) mean enzymatic fluxes during 1 min of ischemic muscle exercise for three model configurations (*A*) control; (*B*) LDH knockout; (*C*) LDH knockout with clamped [NAD] and [NADH] (mean fluxes averaged over 1 min of simulation; ATPase rate = 20 mM/min; OxPhos rate = 0 mM/min). The magnitude of each enzymatic flux is colour-coded between 0 (dark blue) and 16 mM/min (dark red). (*D*) [ATP] dynamics, (*E*) [NADH] dynamics and (*F*) pH dynamics. (*G*) Accumulation ($\Delta[L_{total}] = [L_{total}](t = 1\,\text{min}) - [L_{total}](t = 0)$) of fructose–biphosphate (FBP), glucose-3-phosphate (G3P), pyruvate (PYR) and lactate (LAC) during 1-min ischemic muscle exercise.

interactions of these enzymes with cytoskeletal filament proteins or calcium-binding proteins (Sola-Penna et al., 2010; Singh et al., 2004) that have been proposed to modify glycolytic activity *in vivo* (Schmitz et al., 2013). Moreover, because of a lack of delayed coupling between demand and supply, the lumped OxPhos model might overestimate the observed oscillations.

However, glycolysis has long been known as an ubiquitous example of biochemical oscillatory behaviours (Hess & Plesser, 1979). First identified in yeast (Ghosh & Chance, 1964) and later reproduced in cell-free extracts of skeletal muscle (Tornheim & Lowenstein, 1974), oscillations in glycolytic intermediates, including NADH, triggered by a rapid change in myoplasmic ATP/ADP ratio have also been reported in intact cardiomyocytes (O'Rourke et al., 1994). In this light, it is essential to note that the concept of first-order behaviour of myoplasmic P-metabolite concentrations in response to a step change in ATP turnover originates from computer simulations using an electrical analog model of muscle respiration that excludes glycolysis (Meyer, 1988). Based on these considerations, we conclude that our finding of second-order underdamped behaviour in our computational model of glucose-1-P and ATP metabolism in an FT oxidative skeletal myofibre in response to a step change in ATP turnover represents a generic trait of the ATP metabolic network in mammalian cells.

Thirdly, our simulations contribute to an understanding of the phenomenon of 'aerobic lactate production' in skeletal muscle (Sahlin et al., 1987). This term refers to the energetically seemingly wasteful conversion of pyruvate to lactate, despite presumed adequate oxygen availability and mitochondrial OxPhos capacity, first observed in cancer cells (Warburg, 1956) and later in other cell types, including skeletal myofibres (e.g. Connett et al. (1984); Spriet et al. (2000)). A recent study on cancer cells concluded that insufficient capacity of mitochondrial redox shuttles to buffer elevated glycolytic NADH production drives lactate production in this particular glycolytic cell type and proposed this outcome may be generalizable to other cell types, including skeletal myofibres (Wang et al., 2022). Our findings reject the latter contention: LDH knockout simulations show that oxidative recycling of myoplasmic NADH in the FOG myofibre phenotype in and by itself suffices to maintain redox balance over the aerobic range of ATP turnover rates (Fig. 9), accompanied by faster and steeper on-kinetics of OxPhos (Fig. A.6*A,B*). This outcome is in agreement with empirical findings that patients with muscular LDH deficiency experience no symptoms when exercising at low to moderate intensities (Kanno et al., 1988; Miyajima et al., 1995). Simulations of the intact metabolic model reveal that aerobic lactate production in this myofibre phenotype is driven by the mass action effect of LDH substrate accumulation shortly after activation of glycolysis

by ATP turnover (Fig. A.3). Indeed the majority of lactate produced at low and intermediate ATP turnover rates was predicted to occur in this early time frame of transition to a new steady state (see Fig. A.2). Furthermore the predicted behaviour of the LDH-KO model under ischemia (Fig. 10) is in agreement with studies on LDH deficiency, where patients experience muscle cramping and myoglobinuria (dominantly) induced by ischemic exercise due to an inability to maintain ATP homeostasis (Kanno et al., 1988; Miyajima et al., 1995). Muscle biopsies revealed increased concentrations of pyruvate and glycolytic intermediates (e.g. fructose 1,6-biphosphate (FBP)) after ischemic exercise, while lacking the characteristic lactate accumulation. In particular, observed higher levels of glycerol 3-phosphate (G3P) were attributed to an increased activity of G3PDH. Clamping the myoplasmic redox state, computer simulations indicate that intolerance to high-intensity exercise in LDH-deficient patients is mainly due to the failure of homeostasis in the myoplasmic redox balance rather than the accumulation of glycolytic intermediates in the distal part of the pathway. Taken together, the picture that emerges from these simulations is that aerobic lactate production in FOG myofibres is simply a by-product of metabolic flexibility associated with expressing high levels of LDH, affording continual ATP synthesis under conditions of compromised vascular oxygen supply or high-intensity exercise. A similar view has previously been proposed by others (e.g. Glancy et al., 2021), albeit using qualitative reasoning on the basis of the high (i.e. $K_{app}^{LDH} \approx 10^4$) value of the apparent equilibrium constant of the LDH reaction. The LDH reaction is crucial for maintaining cytoplasmic redox balance under conditions where metabolic demand exceeds the capacity for oxidative ATP synthesis. Note that any lactate produced by FOG myofibres is not lost to the body but made available as oxidative substrate to other cells and tissues, including neighbouring red myofibres (e.g. Ahlborg et al. (1975); Sahlin et al. (2002)). Thus, the phenomenon of aerobic lactate production does not necessarily represent a metabolic inefficiency at the level of the whole organism.

Finally, the presented simulations provide insights into two lingering debates in skeletal muscle biochemistry. Regarding the longstanding debate on whether or not lactate production causes myofibre acidification (e.g. Böning et al., 2005; Kemp et al., 2006; Robergs et al., 2004; Vinnakota and Kushmerick, 2011), the proposed model highlights the often-misconstrued intricacy of myoplasmic proton dynamics. Particularly for analysing measured *in vivo* pH dynamics, it has proven useful to consider the combined effect of ATPase activity and ATP resynthesising pathways (e.g. CK or glycolysis) in stoichiometric equations phenomenologically describing the net contribution to cytoplasmic proton dynamics (e.g. Marcinek et al. (2010)). Based on this approach,

net proton uptake can be observed for ATPase activity balanced by PCr breakdown or OxPhos, and a net proton generation for ATPase flux balanced by glycolytic ATP synthesis with concomitant lactate production. However, lumping together enzyme-catalysed reactions into a single net reaction faces limitations in mechanistically linking observed behaviours and the underlying physiology. Particularly ATP hydrolysis and its simultaneous regeneration through the glycolytic pathway are often assumed to have a total proton stoichiometry factor of $+2$.[1] While this 1:1 ratio of lactate production and glycolytic $H^+$ generation is supported experimentally (Marcinek et al., 2010) as well as by our model in the context of an essentially closed system of (resting) muscle under ischaemia (simulations not shown), it becomes more complex in settings with bigger accumulation of glycolytic intermediates, the presence of oxidative ATP synthesis through cellular respiration of pyruvate or when accounting for the effects of proton coupled mono-carboxylate transport or carbonic anhydrase, even under conditions with overall stable ATP levels in the myoplasm. Using our detailed model, we obtain unique insights into the underlying mechanisms of myoplasmic proton dynamics during a wide range of exercise protocols. At a pH level close to the basal level, ATPase, as a single reaction, predominates in proton generation. However, with falling pH, proton production by ATPase decreases (Fig. A.5) and further generation of free protons is increasingly driven by the glycolytic pathway. Yet the reaction of lactate production catalysed by LDH by itself is not the causality behind a decrease in pH. In fact, GAPDH has the strongest effect on cellular acidification within the glycolytic pathway. However focusing on individual reactions does not necessarily provide meaningful insights into muscle physiology as LDH and PYK counter-act the proton production of other glycolytic reactions considerably, leading to a rather small net acidification (Fig. 7*B*). Similarly in working muscle fibres, free protons are tightly balanced, and the net change in pH is small compared to the proton production fluxes of individual reactions. This showcases that though phenomenological descriptions are useful for a coarse-scale analysis of experimental observations, common interpretations such as lactate production is responsible for myoplasmic acidification, can be misleading and are best avoided without additional context.

Furthermore, LDH KO simulations showed that LDH's role in counteracting acidification is not a necessary pre-requisite for muscular function. During low intensity exercise, a complete absence of LDH can be compensated by increased flux through OxPhos, resulting in a negligible difference in pH dynamics between the control system and LDH KO system (Fig. 8*C*). Crucially although LDH significantly contributes to proton uptake during high intensity exercise, with proton consumption fluxes close to the proton consumption of OxPhos, LDH KO simulations under ischaemia predict that LDH's function in maintaining relatively low cytoplasmic NADH levels is fundamentally more important for maintaining muscle function under compromised oxygenation or high ATP demand than its role in counteracting cellular acidification (see Fig. 10*E*,*F*).

Regarding contentions that lactate is always the product of glycolysis (Glancy et al., 2021; Rogatzki et al., 2015), our model simulations indicate that this is untenable in an absolute sense. Steady-state myoplasmic concentrations of pyruvate and lactate during exercise are both predicted to progressively increase with increasing ATP turnover (Fig. A.3). In fact our model simulations predict that immediately upon a step change in ATP turnover rate, any lactate in the myoplasm or interstitial matrix is converted to pyruvate to fuel mitochondrial ATP production while glycolysis is ramping up (Fig. A.2). This particular model prediction, however, may alternatively reflect the absence of any feed-forward control of glycolytic flux in the present model. Two previous studies of glycolysis in skeletal muscle have proposed that sensing and transduction of the step change in myoplasmic calcium concentration upon myofibre recruitment, which precedes any increase in ATP turnover rate, may indeed be operational *in vivo* (Crowther et al., 2002; Schmitz et al., 2012).

## Modelling muscle energetics

Computational modelling has a rich tradition in muscle energetics. Here, we discuss the proposed model in the context of the existing literature and potential applications.

**Model improvements and comparison to existing models.** Similar to previous models of ATP metabolism in muscle fibres (e.g. Jeneson et al., 2000; Kushmerick, 1998; Vicini & Kushmerick, 2000), the proposed model incorporates the key components of ATP homeostasis, namely ATP turnover by cytoplasmic ATPases and ATP synthesis by CK and OxPhos, which are linked by a feedback control loop. In addition the proposed modeling framework incorporates a detailed model of G1P catabolism including redox coupling to OxPhos, AK buffering of ATP/ADP, LDH buffering of cyto-plasmic redox, lactate exchange with extracellular space and computation of the emergent state variable cyto-plasmic pH building on previous work by Vinnakota and coworkers in their study of resting biochemical state in

---

[1] This refers to the multiauthor commentary on the Point-Counterpoint debate in *J Appl Physiol* 110(5):1493-1496 (2011), https://doi.org/10.1152/japplphysiol.00242.2011.

an inactive muscle fibre (Vinnakota et al., 2006). Thus, the proposed modelling framework sets itself apart from existing computational models of energy metabolism in skeletal muscle that only focus on individual pathways (e.g. Choe et al., 2025; Lambeth & Kushmerick, 2002; Lai et al., 2007; Schmitz et al., 2013; Vinnakota et al., 2006; Wu et al., 2007) as well as previous modelling efforts of both glycolytic and oxidative ATP synthesis that are limited to the resting state (Vinnakota et al., 2010) or dismiss thermodynamic constraints and/or the effects of pH dynamics (e.g. Dash et al., 2007, 2008; Korzeniewski & Liguzinski, 2004). A detailed model of glycolytic and mitochondrial energy metabolism in skeletal muscle with a mixed feed-forward and feed-back control structure of high complexity was previously developed to investigate dynamic behaviour during exercise. However, only comparatively small jumps in ATP demand were considered, and the validity of the dynamic model response was tested against steady-state values (Li et al., 2009).

## Limitations

The presented results are based on a mathematical model capturing the pH dependency and thermodynamics of the simulated biochemical reactions. Nevertheless, the presented model is a simplification of the underlying physiological system. The associated limitations are discussed below.

**Uncertainty of parameters and validation.** The proposed model has a high number of variable model parameters with $N = 85$ (see Tables S2 and S3 in the Supplementary Material). Yet a particular strength of the modelling framework is that most parameters can be fixed based on data from *in vitro* experiments (i.e. a data-rich class of data) together with thermodynamic constraints (i.e. first principles). Nevertheless, there exist a few parameters for which appropriate experimental data are currently not available. These parameters were either obtained by utilising values from measurement conditions slightly deviating from the simulated conditions (e.g. the CK reference equilibrium constant), fitting the model to $^{31}$P-MRS data (see Section 3.1.1) or heuristic parameter adjustments yielding consistency with the expected system behaviour (e.g. the PFK regulation). Particularly, the latter yields parameters subject to considerable uncertainty. Furthermore, the model parameters were obtained from different species (e.g. rat, human, rabbit, mouse and frog) and muscles (e.g. gastrocnemius, gracialis and flexor digitorum superficialis). As enzyme activities vary between species and muscle (fibre) types (Bass et al., 1969), the model's parameterisation was derived from FT mammalian skeletal muscle, if

available. We further note that the model showed the highest sensitivity to changes in the myoplasmic pH (Section 3.1.3). Most maximum enzyme activities were empirically corrected given the current pH value (cf. Vinnakota et al. (2006)); however, such data were not available for all considered reactions.

While the model predicted behaviours, closely replicating observation from $^{31}$P-MRS measurements, direct validation of the predicted internal states (e.g. cytoplasmic redox state and glycolytic intermediates) is currently not possible due to methodological limitations. Yet the *in silico* experiments presented in Figs. 9*B* and 10 showcase that the model is able to qualitatively capture the experimentally observed behaviour, for example, the effect of perturbations in redox balance caused by LDH deficiency.

**Kinetic models.** All enzyme kinetics in the proposed model were adopted from the literature (Connett, 1989; Jeneson et al., 1996; Lambeth & Kushmerick, 2002; Vinnakota et al., 2010; Waser et al., 1983). The majority of enzyme-catalysed reactions are modelled based on a rapid equilibrium assumption for on and off binding-reaction kinetics between metabolites and the enzyme. This assumption might have limitations for flux rates close to the maximum enzyme activities. Furthermore, enzyme-catalysed reactions are assumed to be reversible and thermodynamically constrained, satisfying the Haldane relation. Exceptions are the kinetic descriptions of ATPase, PFK and OxPhos. Their limitations are specifically discussed below.

*ATPase model.* The kinetic description of ATPase is a superposition of a fixed basal rate and an activity-dependent contribution. The simplification of a fixed activity-dependent ATP turnover misses regulatory mechanisms in response to fatiguing exercise. Thus, when simulating extreme scenarios, for example, high-intensity exercise to exhaustion, the model can become numerically unstable. This can potentially be fixed by using a more detailed ATPase model.

*Glycolytic substrates.* We note that energy substrate supply through blood glucose and thus the hexokinase reaction (i.e. converting glucose to G6P) are not considered. Furthermore, due to the lack of a suitable kinetic model, glycogen phosphorylase (GPa and GPb) is not explicitly modelled in the simulated reaction network. Thus, PFK is the first rate-limiting enzyme of the glycolytic pathway. This limits the reliability of the model predictions, particularly for the dynamics of G6P and F6P.

*PFK model.* Despite being intensely studied, there is still a lack of a thermodynamically constrained model of PFK kinetics considering the (coupled) regulatory effects

of multiple metabolites such as ATP, ADP, AMP, Pi and pH dynamics. The utilised kinetic model (Connett & Sahlin, 2010; Waser et al., 1983) describes an irreversible biochemical reaction without thermodynamic restriction. Under physiological conditions, PFK is considered irreversible with $\Delta G'^0_{\text{PFK}} = -15.62\,\text{kJmol}^{-1}$ (Li et al., 2011). Yet the lack of thermodynamic constraints may cause non-physical behaviours, specifically when simulating strong perturbations in ATP balance. Thus, we tested the thermodynamic consistency by calculating the transformed free energy of the PFK reaction. We observe that $\Delta G'_{\text{PFK}}$ remained negative throughout the simulated period, verifying the thermodynamic consistency of the model within the boundary of the considered conditions.

*Mitochondrial model.* The kinetic description of the OxPhos model is based on an empirical sigmoid Hill function with feedback regulation through ADP (Jeneson et al., 1996). We added regulation via inorganic phosphate and pyruvate and found that Pi regulation is necessary to prevent a complete depletion of inorganic phosphate during recovery. All kinetic parameters of the utilised phenomenological model were derived from experimental findings reported in literature, including the constraints for the optimised parameters. Yet this model lacks delayed dynamics between demand and supply. This simplification may potentially lead to an over-estimation of the observed glycolytic oscillations. Deeper insights into the function of mitochondria would require a more detailed OxPhos model (e.g. Beard (2005)). While the proposed model describes an FOG fibre, another limitation of the implemented OxPhos model is the lack of fatty acid oxidation to fuel the resting state.

**Substrate exchange between compartments.** The literature regarding the kinetics of carbon dioxide and lactate transport is sparse. Thus, their implementation is subject to considerable uncertainty. We note that the implemented transport mechanisms are a simplification of mass exchange in real muscle fibres; for example, the selected MCT model discards the symport of non-lactate monocarboxylates, such as pyruvate. Furthermore, $Na^+/H^+$ anti-transport is not included in the model. While $Na^+/H^+$ exchange activity does contribute to overall proton efflux in skeletal muscle fibres (Juel, 1998), lactate/$H^+$ co-transport is considered the most important proton exchanger for pH regulation during muscle exercise due to its markedly higher capacity for proton exchange (Juel, 1997).

Moreover, the presented simulations neglect oxygen dynamics and its regulatory effects on the metabolic network. Compromised oxygen supply to myofibres during high-intensity work is, however, not uncommon (Degens et al., 1998), and oxygen availability is highly dynamic. For example, we do not consider vascular

recruitment during muscle activation (Segal, 2005) or fibre-type-specific capillarisation (Glancy & Balaban, 2021). Finally, note that clamping the extracellular lactate concentration to 1 mM overestimates lactate removal from the extracellular space during high-intensity exercise (e.g. Sahlin et al. (1976)).

## Conclusion and outlook

Within this work, we have developed a mathematical model for the quantitative investigation of energy metabolism in working FOG fibres. Besides measurable metabolic variables, the model can predict the dynamics of variables that have proven challenging to probe experimentally in an intact physiological system, such as the myoplasmic redox state and reaction fluxes. The presented simulations yield novel insights into how reported experimental observations, such as aerobic lactate production in muscle, emerge from the integrative behaviour of the metabolic network and are essential for maintaining robust energy and redox balance across a 100-fold working range. In the future, we plan to replace the phenomenological OxPhos model with a more detailed one. We anticipate that the proposed simulation framework will further enhance our under-standing of muscle fatigue in health and disease, guide the development of treatment strategies for diverse neuro-muscular disorders and improve the interpretation of data from $^{31}$P-MRS during rest, exercise and recovery.

## Appendix

### Appendix

### Extended results

**Correlations of optimised parameters.** To characterise the independence of the estimated model parameters (see Section 2.2.2), we calculated Pearson's correlation coefficient for all pairs of parameters. The correlation coefficients were determined using all sets of parameters $\boldsymbol{q}$ from the 20 optimisation runs with different initial populations. Figure A.1 shows the resulting correlation matrix together with the corresponding *P*-values.

**Experiment 1: Redox contributions, pyruvate and lactate dynamics.** The proposed metabolic model comprises four reactions that contribute to the cytoplasmic redox state. Their total effect on the redox state is summarised in Fig. 5. Figure A.2 shows for three contraction intensities (low, medium and high) the flux rates through each individual reaction contributing to the myoplasmic redox state, that is, the reactions catalysed by GAPDH, G3PDH and LDH as well as OxPhos. The simulations showcase a tightly balanced system of $NAD^+$ reduction and NADH

oxidation, enabling rapid changes in metabolic flux rates (in the range of mM/min), yielding considerably lower total NADH flux rates in the micro Molar per minute range (see Fig. A.2*D–F*). Notably, the redox potential reaches a steady state in less than 2 min after the onset of muscle activation, although several reactions still show transient behaviour, for example, the G3PDH-reaction.

Furthermore, the intracellular lactate and pyruvate concentration dynamics are shown in Fig. A.3 for the same simulation protocol. Lactate accumulation, as well as a comparatively small accumulation of pyruvate, is observed for all three simulated contraction intensities, including low intensity exercise, with ATPase rates below the maximal capacity for oxidative ATP production (Fig. A.3*A*). Accumulation of both lactate and pyruvate facilitates increased pyruvate consumption through OxPhos. Thus, in a biochemical network including LDH, lactate accumulation during low-intensity exercise is a necessary prerequisite to steer pyruvate uptake in the direction of OxPhos through increased levels of PYR.

**Experiment 1: Transformed Gibbs energy of the redox half reaction.** The biochemical reaction of the redox half reaction is given by:

$$NAD^-_{ox+2e} \rightleftharpoons NADH_{red} . \qquad (A.1)$$

The associated reference reaction is defined as:

$$NAD^- + H^+ + 2e^- \rightleftharpoons NADH^{2-}, \quad n^{redox} = -1 . \quad (A.2)$$

The redox half-reaction is not a stand-alone reaction in the model. Thus, the apparent equilibrium constant necessary for the calculation of the transformed Gibbs energy of the biochemical reaction (see eqn. (11)) is not

### Table A.1. Properties of the reference species of the redox half reaction.

| Reference species | $\Delta_f G^0$ (kJ/mol)[a] | $v_i$ | $z_i$ |
|---|---|---|---|
| $H^+$ | 0 | −1 | 1 |
| $NAD^-$ | 0 | −1 | −1 |
| $NADH^{2-}$ | 22.65 | 1 | −2 |

[a]Values were taken from the BISEN database (Vanlier et al., 2009) (see also Alberty, 2003). The values are determined at 25°C, 1 Bar and $I = 0$.

explicitly computed as part of the model. Instead, the value was calculated separately using the same method as implemented in BISEN. We used the standard Gibbs energies of formation listed in the BISEN database (see Table A.1).

The standard Gibbs energy of the reference reaction is the difference in the energies of formation of the reactants and products:

$$\Delta_r G^0_k = \sum_j v^k_j \, \Delta_f G^0_j . \qquad (A.3)$$

The Gibbs energies of formation of the reference species in Table A.1 were reported at zero ionic strength. The standard Gibbs energy of the reference reaction was empirically adjusted to the ionic strength in the model using the following equations:

$$\Delta_r G^0_k = \Delta_r G^0_k (I = 0) - \beta_G(T, I) \sum v_i z_i^2 , \quad (A.4a)$$

$$\beta_G = \frac{\alpha_G(T) \, I^{1/2}}{1 + 1.6 \, I^{1/2}} , \qquad (A.4b)$$

$$\alpha_G = 9.20483 \cdot 10^{-3} T - 1.28467 \cdot 10^{-5} T$$
$$+ 4.95199 \cdot 10^{-8} T, \qquad (A.4c)$$

with $I = 0.1\,M$ and $T = 298.15\,K$ representing the ionic strength and temperature used in the presented simulations. The stoichiometry factors, $v_i$, and charges, $z_i$, of the reference species are listed in Table A.1. We note that the BISEN database does not include any enthalpies of formation necessary for adjusting $\Delta_r G^0_k$ to the temperature in the model, even though literature values are available (Alberty, 2003). We decided to forgo the empirical temperature correction to stay consistent with the rest of the model.

The equilibrium constant of the reference reaction was calculated (eqn. (10)) to compute the apparent equilibrium constant for the associated biochemical reaction using the simulated proton dynamics (eqn. (9)). The dissociation constant for the binding polynomials (eqn. (4)) is not specified in the BISEN database for the

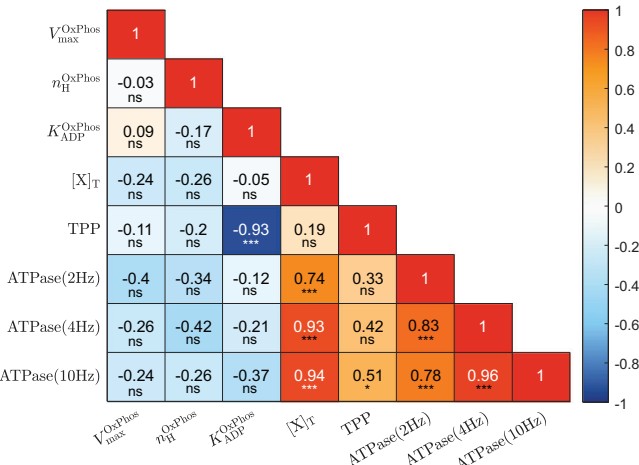

**Figure A.1. Correlation matrix of the optimised parameter values**
The significance of the correlation is indicated by symbols (non-significant (ns): *P*-value $\geqq$ 0.05, * : *P*-value < 0.05, ** : *P*-value < 0.01 and *** : *P*-value < 0.001).

metabolites involved and, therefore, were assumed to be infinite (no binding).

Finally, $K_{app}$ was used to determine the dynamics of the transformed Gibbs energy of the redox half reaction (eqns. (12) and (11)).

**Experiment 1: Contribution to proton generation.** The proposed model quantifies the proton flux of all biochemical reactions in the system, while accounting for the effects of the concomitant changes in pH on reaction kinetics and thermodynamics. The proton generation fluxes of a selection of biochemical reactions

and for three contraction intensities (low, medium and high) are summarised in Fig. A.4. Although the proton generation flux of individual pathways and reactions is highly dynamic, a net proton generation is observed for all three exercise intensities after roughly 30 s of exercise. Figure A.4*A*,*D* and *G* shows that the initial alkalinisation during muscle exercise can be attributed to ATPase activity being balanced by PCr breakdown and the subsequent myoplasmic acidification to balance ATP hydrolysis by glycolytic ATP synthesis. Notably, the main proton generation flux is caused by ATP hydrolysis (ATPase) itself (Fig. A.4*B*,*E*, and *H*). However, a decreasing pH value

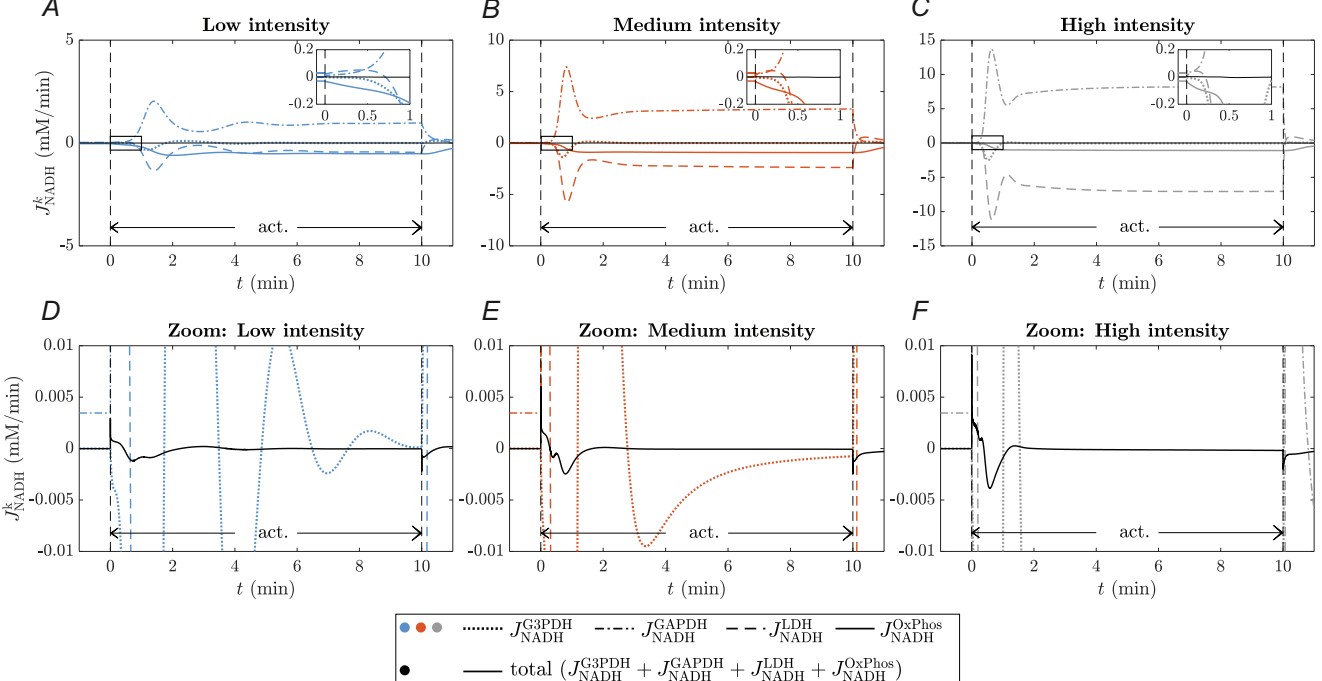

**Figure A.2. Individual enzyme contributions to myoplasmic redox balance**
NADH flux rate ($J^k_{NADH}$) through each associated (enzyme-catalysed) reaction for three different contraction intensities (Blue: ATPase rate 10 mM/min, Orange: ATPase rate 20 mM/min, Grey: 30 mM/min). A positive flux indicates NADH generation; a negative flux indicates NADH oxidation.

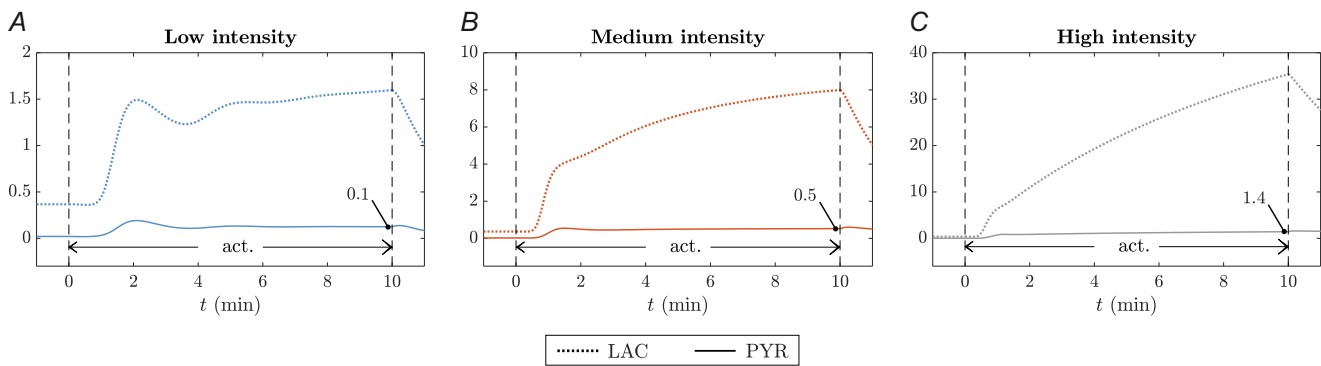

**Figure A.3. Pyruvate and lactate dynamics depending on the exercise intensity**
Pyruvate and lactate time course during 10 min muscle exercise for three different contraction intensities, that is, ATPase rate 10 mM/min (*A*), ATPase rate 20 mM/min (*B*), ATPase rate 30 mM/min (*C*).

causes a reduced proton stoichiometry factor in the ATP hydrolysis reaction (see Fig. A.5*A*) and hence, the relative proton generation contribution of ATPase and glycolysis dynamically evolves over time. This behaviour is observed at all considered workloads and is most pronounced at the highest simulated workload. Apart from the transient alkalinisation resulting from the rapid breakdown of PCr by CK, proton consumption is primarily driven by OxPhos. Similar to ATPase, its contribution to proton balance decreases with falling pH (Fig. A.5*D*), while an increasing proton uptake by

MCT can be observed over time. Notably, the system is tightly balanced, resulting in a relatively small generation of free protons (Fig. A.4*A*,*D* and *G*, black dashed line) compared to the total proton production flux within the system. Similarly, the total proton production within the glycolytic pathway, mainly driven by GAPDH and PGK, widely exceeds the net proton production of glycolysis due to counteracting proton consumption by, for example, LDH and PYK (Fig. A.4*C*,*F* and *I*). Figure A.4 shows a selection of six glycolytic enzymes with the highest absolute proton fluxes within the pathway. The

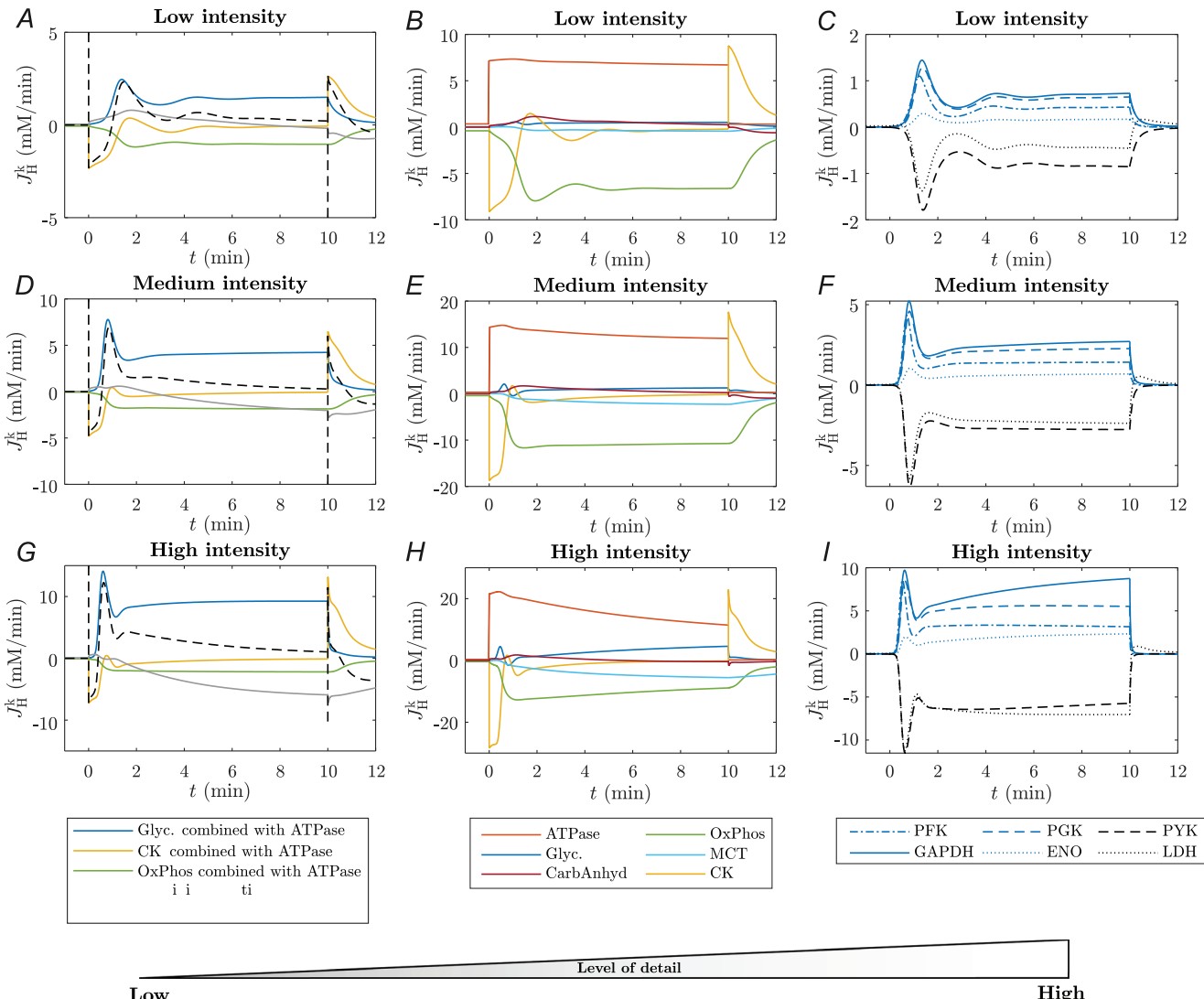

**Figure A.4. Multiscale analysis of dynamic myoplasmic proton balancing**
Proton production fluxes ($J_H^k$) during 10 min of exercise at three different contraction intensities (Low: ATPase rate 10 mM/min, Medium: ATPase rate 20 mM/min, High: 30 mM/min). A positive flux indicates proton generation; a negative flux indicates proton consumption. (*A*, *D*, *G*) proton flux of lumped pathways, that is, the glycolytic pathway combined with the fraction of ATPase balanced by glycolytic ATP synthesis, and OxPhos combined with the fraction of ATPase balanced by oxidative phosphorylation. The dashed line indicates net proton flux in the system, that is, the sum of all proton fluxes in the myoplasm. (*B*, *E*, *H*) proton flux of individual metabolic pathways. AK and $CO_2$ diffusion have a proton flux near zero and are not shown. (*C*, *F*, *I*) breakdown of the contribution to the proton production by a selection of glycolytic enzymes.

corresponding proton generation stoichiometries plotted over pH are summarised in Fig. A.5C. With falling pH, an increasing stoichiometry coefficient of GAPDH and PYK can be observed, which is counteracted to some degree by a decreasing stoichiometry coefficient of PFK and PGK.

Apart from pH, the proton stoichiometry coefficient also depends on the concentration of metal ions. In detail while $[K^+]$ is clamped at 120 mM, $[Mg^{2+}]$ is variable. However, the change in the concentration of free magnesium ions stays within a range ($1.0\,\text{mM} \leq [Mg^{2+}] \leq 1.2\,\text{mM}$) that has a negligible effect on the stoichiometry coefficient values during the three considered workloads. Thus, Fig. A.5 shows the proton generation stoichiometry coefficient for basal $[Mg^{2+}]$.

**Experiment 2: Redox contributions, pyruvate and lactate dynamics.** The conducted *in silico* experiment 2 compares the behaviour of the control model and the LDH-KO model given a 20-min-long exercise protocol with ATPase rates up to the maximal rate of oxidative ATP production ($J_{\text{ATP,max}}^{\text{OxPhos}} = 20\,\text{mM/min}$). Figure A.6–*A,B* exemplarily shows the NADH flux rates through the reactions catalysed by GAPDH, G3PDH and LDH as well as OxPhos during low intensity and medium intensity exercise. Lactate and pyruvate dynamics for the same simulation protocol are shown in Fig. A.6,*C,D*.

Figure A.7 (A and D) compares the NADH time course of the control model and the LDH-KO model

for a selection of simulated ATPase rates ranging from 0.5 to 20 mM/min. The simulation results suggest that LDH activity dampens oscillations in NADH content during jumps in ATP demand at ATPase rates. The transient limited availability of myoplasmic NADH in the LDH KO system at some of the simulated exercise intensities results in a decrease in myoplasmic NADH utilisation by the aerobic pathway (Fig. A.7*C,F*) and thus, a reduced capacity for oxidative ATP production (Fig. A.7*B,E*). The observed change in the ATP ($s_{\text{ATP}}$) and NADH ($s_{\text{NADH}}$) stoichiometry factor of the OxPhos model is very small; however, it is necessary to avoid complete NADH depletion and to ensure mass conservation.

**Underdamped dynamic behaviour.** The model predicts a dynamic behaviour similar to a second-order under-damped system in transitions between steady states, prominently at low ATP turnover rates. This behaviour is manifested, for example, in the time course of myoplasmic PCr and NADH concentrations (Fig. A.8*A,D*), reflecting the oscillating flux rates of the biochemical reactions in the metabolic network. Figure A.8*G* exemplarily shows the metabolic fluxes associated with NADH concentration in the muscle fibre. To investigate the underlying mechanisms of this behaviour, we simulated the dynamic response to 10 min of low intensity exercise (ATPase rate: 10 mM/min) of the control model, a

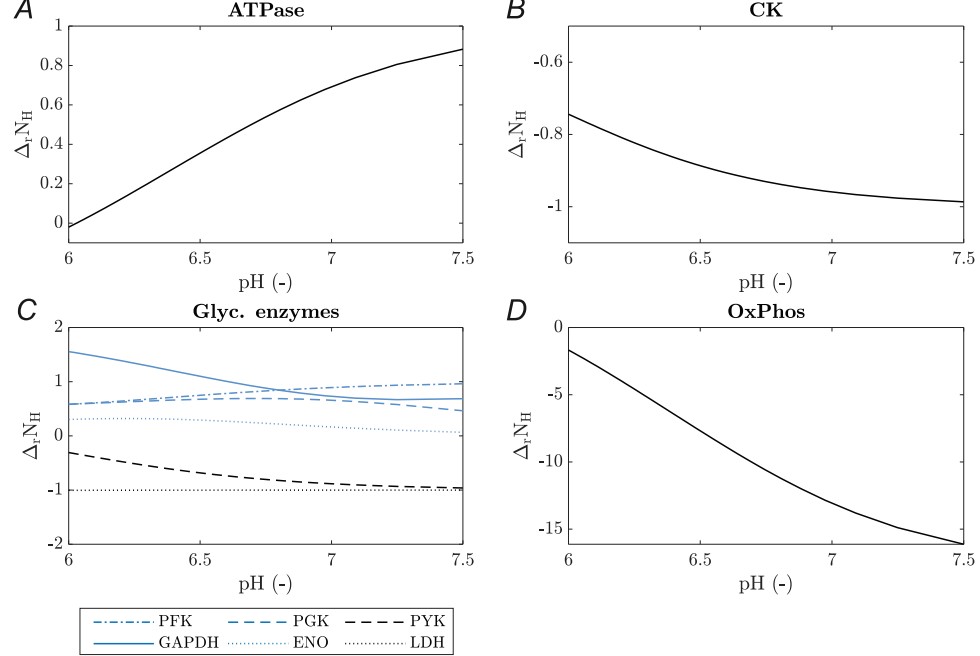

**Figure A.5. pH-dependent proton generation stoichiometries**
(*A*) ATPase, (*B*) CK, (*C*) a selection of enzyme-catalysed reactions in the glycolytic pathway, and (*D*) OxPhos. A positive value indicates proton generation; a negative value indicates proton consumption. $[Mg^{2+}] = 1.06\,\text{mM}$ and $[K^+] = 120\,\text{mM}$.

model configuration with increased $V_{max}^{OxPhos}$ and a model configuration with increased $V_{max}^{G3PDH}$. Increasing $V_{max}^{OxPhos}$ to 1.5 times the control value enhances the underdamped behaviour of the system with oscillations lasting throughout the entire 10 min of muscle activation

(Fig. A.8*B,E,H*). In contrast, increasing $V_{max}^{G3PDH}$ to double its control value dampens the oscillations in metabolic fluxes and, thus, the myoplasmic PCr and NADH concentrations slightly compared to the control system (Fig. A.8*C,F,I*).

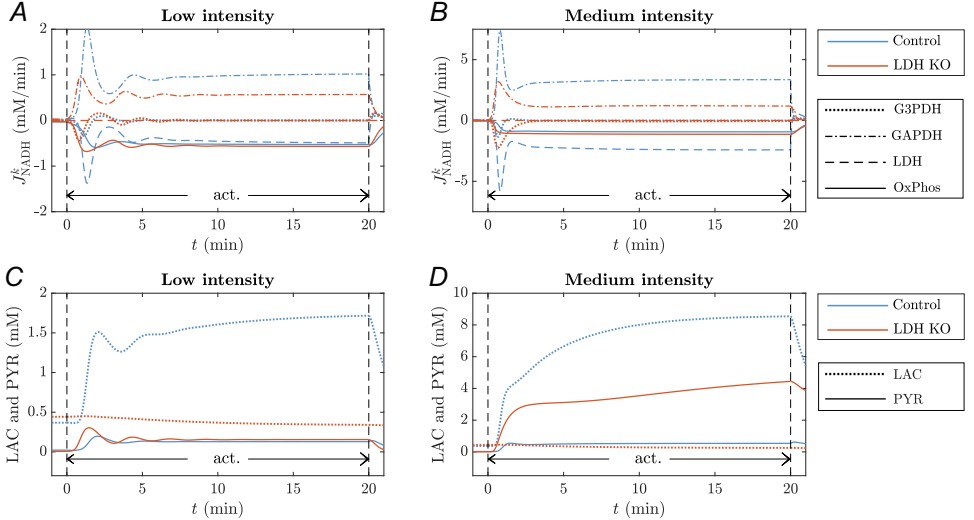

**Figure A.6. Redox dynamics in the LDH-KO system**
Comparison of the control system (blue) and LDH KO system (orange) during 20-min muscle exercise at two different contraction intensities (low intensity: ATPase rate 10 mM/min, medium intensity: ATPase rate 20 mM/min). (*A–B*) NADH flux ($J_{NADH}^{k}$) through each associated (enzyme-catalysed) reaction. A positive flux indicates NADH generation; a negative flux indicates NADH oxidation. (*C–D*) Pyruvate and lactate time course.

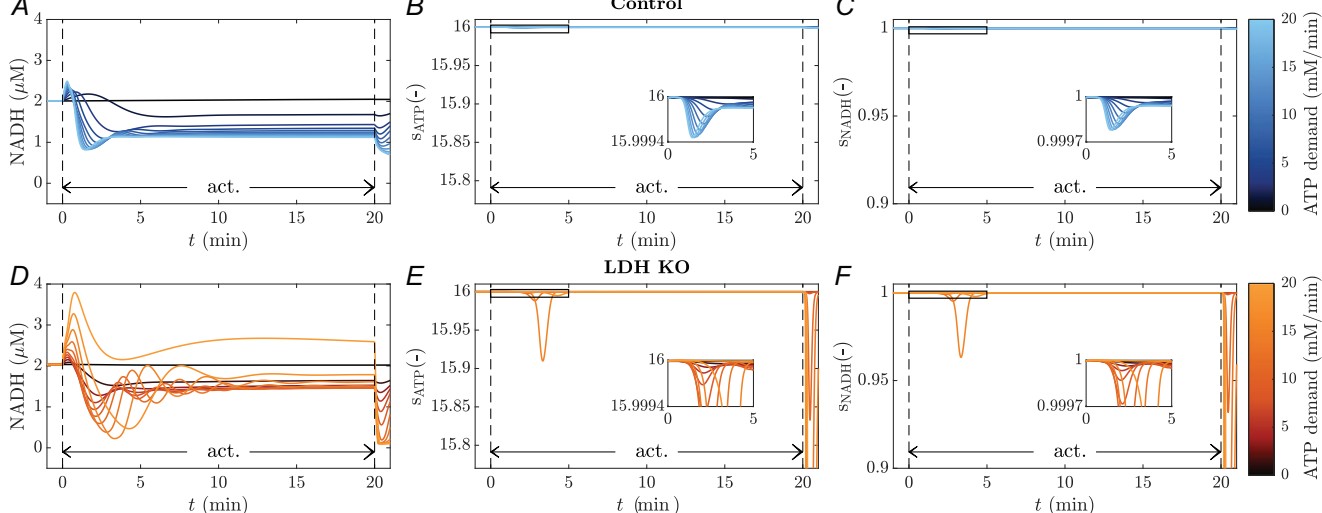

**Figure A.7. Dynamics of the OxPhos stoichiometry factors**
Comparison of (*A–C*) the control system and (*D–F*) the LDH-KO system during 20-min muscle exercise at a selection of 12 ATPase rates ranging from 0.5 to 20 mM/min. The ATPase rate is indicated by the colour gradient. (*A* and *D*) NADH dynamics. (*B* and *E*) Dynamic change in the OxPhos stoichiometry factor for ATP; (*C* and *F*) Dynamic change in the OxPhos stoichiometry factor for NADH.

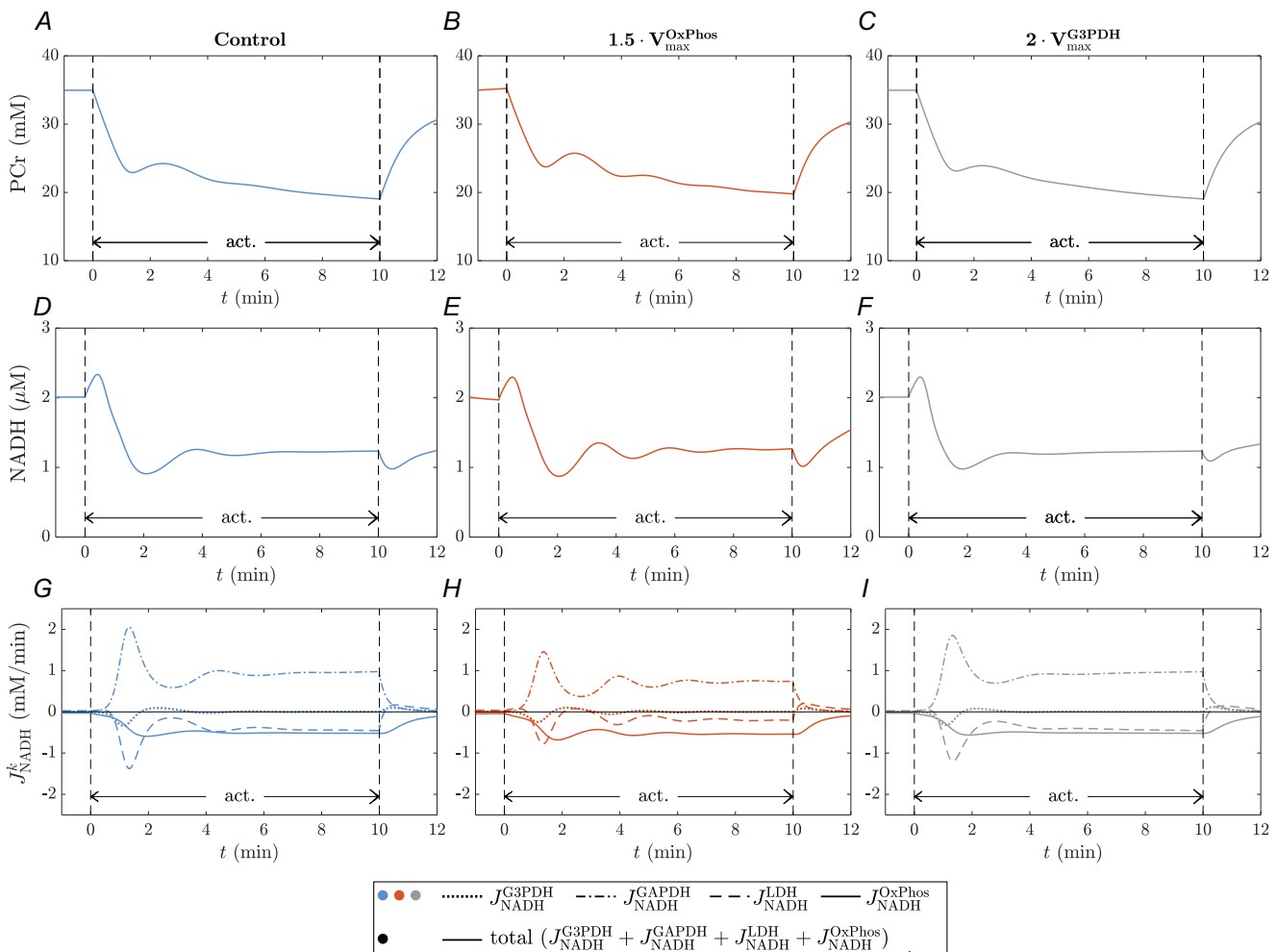

**Figure A.8. Second-order underdamped dynamic behaviour**
Comparison of the dynamic response of the control model (left), a model configuration with increased $V_{max}^{OxPhos}$ (middle) and a model configuration with increased $V_{max}^{G3PDH}$ (right) to 10 min of muscle activation at an ATPase rate of 10 mM/min. (*A–C*) PCr concentration. (*D–F*) NADH concentration. (*G–I*) NADH flux ($J_{NADH}^{k}$) through each associated (enzyme-catalysed) reaction. A positive flux indicates NADH generation; a negative flux indicates NADH oxidation.

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

# Additional information

## Data availability statement

The source code and data used to generate the results and analyses in this study are openly available at the following URL: https://github.com/j-disch/EnergyMetabolism_FOG.git.

## Competing interests

The authors declare that they have no conflict of interest.

## Author contributions

J.D.: Conceptualization; methodology and formal analysis; investigation, software, visualisation and data curation; writing – original draft; writing – review and editing. J.A.L.J.: Conceptualization; methodology and formal analysis; writing – original draft; writing – review and editing; funding acquisition and supervision. D.A.B.: Conceptualization; methodology and formal analysis; writing – review and editing; funding acquisition and supervision. O.R.: Conceptualization; writing – review and editing; funding acquisition and supervision; resources. T.K.: Conceptualization; methodology and formal analysis; writing – original draft; writing – review and editing; funding acquisition and supervision; project administration. All authors have approved the final version of the manuscript and agree to be accountable for all aspects of the work. All persons designated as authors qualify for authorship, and all those who qualify for authorship are listed.

## Funding

This research was funded by the German Research Foundation (Deutsche Forschungsgemeinschaft, DFG) as part of the cluster of excellence 'Data-integrated simulation science (SimTech)' (EXC 2075 390740016) and the priority program SPP 2311 (465195108 and 548605919), the European Research Council (ERC) through ERC-AdG 'qMOTION' (101055186), the National Institutes of Health (NIH) through grants HL173346 and HL154624, and by Stichting Spieren voor Spieren of the Netherlands.

## Acknowledgements

The authors thank Hans Westerhoff for stimulating discussions on glycolytic oscillations.

## Keywords

aerobic lactate production, ATP metabolism, computer simulation, LDH knockout, magnetic resonance spectroscopy, mathematical model, muscle fatigue, skeletal muscle

# Supporting information

Additional supporting information can be found online in the Supporting Information section at the end of the HTML view of the article. Supporting information files available:

**Peer Review History**
**Data S1**

