## [Peer Review History · The Journal of Physiology]

Dynamic balance of myoplasmic energetics, redox state and protons in a fast-twitch oxidative glycolytic skeletal muscle fiber

Jana Disch, Jeroen A.L. Jeneson, Daniel A Beard, Oliver Röhrle, and Thomas Klotz
DOI: 10.1113/JP289702

Corresponding author(s): Thomas Klotz (thomas.klotz@imsb.uni-stuttgart.de)

The following individual(s) involved in review of this submission have agreed to reveal their identity: Gwenael Layec (Referee #3)

Review Timeline:	Submission Date:	11-Jul-2025
	Editorial Decision:	19-Aug-2025
	Revision Received:	26-Nov-2025
	Accepted:	12-Dec-2025

Senior Editor: Bettina Mittendorfer

Reviewing Editor: Christopher Sundberg

Transaction Report:

Dear Dr Klotz,

Re: JP-RP-2025-289702 "Dynamic balance of myoplasmic energetics and redox state in a fast-twitch oxidative glycolytic skeletal muscle fiber" by Jana Disch, Jeroen A.L. Jeneson, Daniel A Beard, Oliver Röhrle, and Thomas Klotz

Thank you for submitting your manuscript to The Journal of Physiology. It has been assessed by a Reviewing Editor and by 2 expert referees and we are pleased to tell you that it is potentially acceptable for publication following satisfactory major revision.

REVISION CHECKLIST:

Please upload two versions of your manuscript text: one with all relevant changes highlighted and one clean version with no

changes tracked. The manuscript file should include all tables and figure legends, but each figure/graph should be uploaded as separate, high-resolution files.

We look forward to receiving your revised submission.

Yours sincerely,

Bettina Mittendorfer
Senior Editor
The Journal of Physiology

REQUIRED ITEMS

- 1) - Author photo and profile. First or joint first authors are asked to provide a short biography (no more than 100 words for one author or 150 words in total for joint first authors) and a portrait photograph. These should be uploaded and clearly labelled together in a Word document with the revised version of the manuscript. See Information for Authors for further details.
- 2) - The reference list must be in alphabetical order, rather than numbered, to comply with our Journal format.
- 3) - Your manuscript must include a complete Additional Information section, including competing interests; funding; author contributions and acknowledgements.
- 4) - Please upload separate high-quality figure files via the submission form.
- 5) - Please ensure that any tables are editable and in Word format, and wherever possible, embedded in the article file itself.
- 6) - Please ensure that the Article File you upload is a Word file.
- 7) - Please include an Abstract Figure file, as well as the Figure Legend text within the main article file. The Abstract Figure is a piece of artwork designed to give readers an immediate understanding of the research and should summarise the main conclusions. If possible, the image should be easily 'readable' from left to right or top to bottom. It should show the physiological relevance of the manuscript so readers can assess the importance and content of its findings. Abstract Figures should not merely recapitulate other figures in the manuscript. Please try to keep the diagram as simple as possible and without superfluous information that may distract from the main conclusion(s). Abstract Figures must be provided by authors no later than the revised manuscript stage and should be uploaded as a separate file during online submission labelled as File Type 'Abstract Figure'. Please also ensure that you include the figure legend in the main article file. All Abstract Figures should be created using BioRender. Authors should use The Journal's premium BioRender account to export high-resolution images. Details on how to use and access the premium account are included as part of this email.
- 8) - Please ensure that all figures and tables have a title and legend, and that they have been cited within the main article text.

EDITOR COMMENTS

Reviewing Editor:

Thank you for submitting your manuscript to The Journal of Physiology. Two expert reviewers evaluated your manuscript

and while both were highly enthusiastic about the comprehensive mathematical modeling approach and potential impact of the work, they both had several points that need to be carefully addressed. Specifically, the model is not able to accurately predict alterations in PCr, pH, and Pi during exercise, which requires more thorough discussion, and there are several additional connections that can be made to the prevailing literature that will help bolster the potential impact. The authors also must adhere to the journal policies regarding reporting of data as supplementary information before a final decision on acceptance can be made.

Senior Editor:

Two expert reviewers and the reviewing editor expressed considerable enthusiasm for this paper and I concur. The reviewers offer constructive feedback to help further improve this high-quality paper. The major points that are important to address are 1) the model's inability to accurately predict alterations in PCr, pH, and Pi during exercise (Reviewer 1) and 2) there are several additional connections that can be made to the prevailing literature that would help bolster the potential impact (Reviewer 2).

Please work with the editorial staff to accommodate the extensive supplemental material in a way that complies with journal guidelines.

REFeree COMMENTS

Referee #1:

This study by Jana et al. evaluates the validity of a computational model of glycolysis and mitochondrial oxidative phosphorylation in fast twitch oxidative glycolytic muscle fibers. Predictions of the model were tested different workload, against experimental data, or conditions simulating enzymatic deficiency (LDH KO). The main findings of this study are: 1) a feedback-driven biochemical model (i.e. not including feed-forward signal from Ca²⁺) can account for the ATP supply-demand relationship over a 100 fold range of ATP demands, 2) LDH plays a fundamental role in NADH balance, and 3) LDH conversion of pyruvate into lactate has little influence on acid-base balance in the muscle.

This is a very comprehensive model of metabolic control of the skeletal muscle that builds on previously developed models from Kushmerick's group. The novelty of this study is to integrate redox regulation of NADH/NAD, which adds significantly to the existing literature and provides novel insights about the mechanism of metabolic control during exercise that would otherwise be difficult to test experimentally in humans/animals. I have a few comments I would like the authors to address.

Main Comments

- While overall quantitatively sound, the models failed to predict the kinetics of PCr, Pi, and pH for the experimental data (Figure 1). For instance, at 2 and 4Hz could be observed in the experimental data and a plateau in the concentration of PCr or Pi could be expected but was not fitted by the model. In fact, a second component for PCr and Pi was predicted, which is inconsistent with the monoexponential response typically observed experimentally. pH was also poorly fitted (2 and 3). Please comment.
- Figure 5A is also not consistent with the monoexponential pattern observed during light to moderate exercise intensities. How does this impact the conclusion of the model about H⁺ handling and NADH ?
- K_{ck} calculation: Golding 1995 (<https://pubmed.ncbi.nlm.nih.gov/7636446/>) provided an equation to K for different conditions of pH and Mg that may better suit the present conditions and improve the accuracy of the model as K_{ck} can vary 10 fold within a pH range of 6-7.2.
- Page 25: the authors state LDH conversion of pyruvate into lactate has little influence on cellular acid-base balance. I would recommend expanding on this point and adding more results on this specific topic. Is the process of lactate production through glycolysis responsible for muscle acidosis? Can the authors identify some of the key contributors to H⁺ production ? if pyruvate conversion into lactate is ruled out (cf. present data), what is the contribution of GAPDH or ATP hydrolysis to H⁺ balance. The manuscript would benefit from a figure similar to figure 8 with the H⁺ production/consumption from GAPDH, ATP hydrolysis and pyruvate conversion into lactate. From the model description, it looks like GAPDH was including in the H⁺ balance calculation to run the model.

Using their model, can the authors provide insights on how to reconcile the discrepancy between experimental data are consistent with glycolysis contributing to muscle acidosis (<https://pubmed.ncbi.nlm.nih.gov/19066935/>, <https://pubmed.ncbi.nlm.nih.gov/20133437/>), whereas computer models seem to refute this theory (<https://pubmed.ncbi.nlm.nih.gov/20308252/>).

Minor comments:

Introduction page 2: in the context of the skeletal muscle and dynamic muscle contraction, the term fermentation seems

inappropriate. I recommend using substrate-level phosphorylation.

Referee #2:

This paper describes a rigorous computational approach to an important physiological question, The relationship to earlier work is clearly explained, and the limitations are fairly stated.

My comments, are largely about further connections which could usefully made to earlier work.

1. Mitochondrial regulation

A major simplification is obviously the approximation used for oxidative phosphorylation. As the authors are aware, there is lot of evidence, largely ³¹P MRS-based, that this relationship holds in general in steady-state aerobic exercise, and in recovery from exercise of a wider range of intensities; it is also consistent with detailed computational-modelling of the whole pathway; and there is quite a large literature on to what extent this relationship is invariant e.g across exercise intensities, and a long debate whether there are or aren't significant additional feed-forward mechanisms. The component reactions are dealt with in some detail in the Appendix, but it's not entirely clear to me how closely the main findings depend on this kinetic approximation, given the importance of this component to overall energy production. This limitation is of course discussed in lines 730-736, although I would say the problem is slightly larger than 'the utilized phenomenological model provides limited insights into the function of mitochondria' - it also has some bearing on at least the robustness of the results about glycolysis.

2. Overall relationships of key rates and metabolites.

It would be interesting to see both glycolytic and oxidative ATP synthesis rates plotted against the concentrations of the MRS-visible metabolites that have attracted attention as potential closed-loop feedback signals. For oxidative ATP the relationship to [ADP] would obviously be close to the model equation, and the general constraints of the CK equilibrium would no doubt make its relationship to [Pi] near-linear, and that to DeltaG, sigmoid in the expected way (cf Fig 6B, which plots this for total ATP demand). But how does glycolytic ATP synthesis rate look on the same plots? This is not just idle curiosity, as it would throw some useful light on what to expect from ³¹P MRS data.

In the same spirit, it would be interesting to know see the relationship between total H⁺ efflux and pH, given the (admittedly inferential) information on this which can be obtained from post-exercise pH recovery kinetics. In this model 'H⁺ efflux' must be dominated by lactate/H⁺ transport, a modelled function of intracellular lactate concentration, modified by transmembrane fluxes of bicarbonate (or CO₂ fluxes amounting to that). It seems that the Na⁺/H⁺ antiporter is not part of the model - what differences does that make? (I think this deserves a comment in the Discussion, at least). I'm aware that this touches on some fundamental aspects of muscle H⁺ physiology that (even after the apparent resolution of the confusions and misunderstandings detailed in Refs 67-71) are not that well understood. I am just suggesting that the H⁺ efflux to pH relationship would make an interesting comparison with previous ³¹P work.

3. Buffering

The computational approach taken here is more fundamental and rigorous than what one might call traditional approaches to cytosolic acid base balance. The latter, however, are still useful in quantitative (you might prefer semi-quantitative) analysis of ³¹P MRS experiments. On a technical level, it would be interesting to know whether the analysis here implies a stoichiometric coefficient for the Lohmann reaction more or less that same as in the detailed analysis of Kushmerick (1997) *Am J Physiol* 272:C1739-47, rather than the larger coefficient of the simpler analysis used in the much of the earlier literature (and still occasionally now).

4. Creatine kinase

How close is CK to equilibrium through all this? This is a useful technical question for ³¹P MRS. Is it close enough for the traditional MRS calculation of [ADP] throughout?. The same question could be asked about AMPK and [AMP].

5. Off-kinetics

The fit to post-exercise recovery kinetics of PCr and pH (where the measurement is possible) is quite good. I'm assuming that lactate production rapidly declines after the end of exercise, otherwise pH would keep going down for longer (depending obviously on the balance between this, H⁺ generation by PCr resynthesis and H⁺ efflux by various means). I would welcome a bit more discussion of this point. As you allude to, although solid evidence is hard to obtain, the rapid shutdown of lactate production, and the consequent reliance on oxidative ATP synthesis to fund PCr restoration even after significantly 'glycolytic' exercise, is sometimes taken as evidence of some sort of feed-forward control mechanism of the sort that the present analysis shows is not necessary to explain responses during exercise. It would be interesting to see a full time-course (exercise and recovery) showing both glycolytic and oxidative ATP synthesis rate - and also, actually, of lactate efflux and total net H⁺ efflux, as per my comment above.

Minor points:

Fig 6A. I don't quite understand the (so to speak) blanks in the 'CK' block?

Line 24 - This confounds two things worth distinguishing: the lower ATP yield of glycolysis to lactate, and loss of lactate from the cell.

Line 175 - not 'acetic' acid.

END OF COMMENTS

**Response to the reviewers:
"Dynamic balance of myoplasmic
energetics, redox state and protons in a
fast-twitch oxidative glycolytic skeletal
muscle fiber"**

November 2025

We thank the reviewers and the editorial team for their constructive feedback, which substantially improved the proposed manuscript. A detailed point-by-point response to the comments raised by the reviewers is given below. Within the revised manuscript, major changes are highlighted in red font.

Reviewing Editor

Thank you for submitting your manuscript to The Journal of Physiology. Two expert reviewers evaluated your manuscript and while both were highly enthusiastic about the comprehensive mathematical modeling approach and potential impact of the work, they both had several points that need to be carefully addressed. Specifically, the model is not able to accurately predict alterations in PCr, pH, and Pi during exercise, which requires more thorough discussion, and there are several additional connections that can be made to the prevailing literature that will help bolster the potential impact. The authors also must adhere to the journal policies regarding reporting of data as supplementary information before a final decision on acceptance can be made.

Senior Editor

Two expert reviewers and the reviewing editor expressed considerable enthusiasm for this paper and I concur. The reviewers offer constructive feedback to help further improve this high-quality paper. The major points that are important to address are 1) the model's inability to accurately predict alterations in PCr, pH, and Pi during exercise (Reviewer 1) and 2) there are several additional connections that can be made to the prevailing literature that would help

bolster the potential impact (Reviewer 2).

Please work with the editorial staff to accommodate the extensive supplemental material in a way that complies with journal guidelines.

Reviewer 1

This study by Jana et al. evaluates the validity of a computational model of glycolysis and mitochondrial oxidative phosphorylation in fast twitch oxidative glycolytic muscle fibers. Predictions of the model were tested different workload, against experimental data, or conditions simulating enzymatic deficiency (LDH KO). The main findings of this study are: 1) a feedback-driven biochemical model (i.e. not including feed-forward signal from Ca^{2+}) can account for the ATP supply-demand relationship over a 100 fold range of ATP demands, 2) LDH plays a fundamental role in NADH balance, and 3) LDH conversion of pyruvate into lactate has little influence on acid-base balance in the muscle.

This is a very comprehensive model of metabolic control of the skeletal muscle that builds on previously developed models from Kushmerick's group. The novelty of this study is to integrate redox regulation of NADH/NAD, which adds significantly to the existing literature and provides novel insights about the mechanism of metabolic control during exercise that would otherwise be difficult to test experimentally in humans/animals. I have a few comments I would like the authors to address.

Major comments:

1. While overall quantitatively sound, the models failed to predict the kinetics of PCr, Pi, and pH for the experimental data (Figure 1). For instance, at 2 and 4Hz could be observed in the experimental data and a plateau in the concentration of PCr or Pi could be expected but was not fitted by the model. In fact, a second component for PCr and Pi was predicted, which is inconsistent with the monoexponential response typically observed experimentally. pH was also poorly fitted (2 and 3). Please comment.

Response: The objective of Fig. 2 is to showcase the performance of the model to capture reported dynamics in PCr, Pi and pH levels in rat hindlimb muscle composed predominantly of FT oxidative fibers after fitting of five model parameters (out of 85 adjustable parameters, excluding the 13 apparent equilibrium constants and the parameters of the empirical functions used to describe the pH dependency of enzyme activities) as well as the ATPase rates (see Table 2). We note that fitting additional parameters may further decrease the discrepancy between data and model predictions; however, we selected this approach to highlight the robustness of the underlying thermodynamic-constraint modeling framework.

Additionally, the model assumes a steady ATPase rate during muscle activation, disregarding any dynamic adaptation due to muscle fatigue that is likely to play a role at high stimulation frequencies. We further note that the accuracy of the experimental data is approximately ± 1 mM and ± 0.1 pH units. Based on these considerations, we would rate the overall quality of the model predictions as solid. In the following, we will briefly discuss the discrepancies between the model and the experimental data (particularly for the pH dynamics of the 2 Hz stimulation) and the considerations raised by the reviewer.

Both the experimental data and model predictions deviate from a mono-exponential behaviour; however, compared to the experimental data for the 4 and 10 Hz stimulation, the model underestimates the initial increase in Pi. The mono-exponential approximation was originally proposed in the classic paper from Meyer (1988), ‘A linear model of muscle respiration explains monoexponential phosphocreatine changes’, based on a non-mechanistic electrical analogue circuit model of muscular ATP balance lacking glycolysis. Yet, existing literature suggests that (un)dampened oscillations of glycolytic intermediates and associated metabolites, such as NADH, in response to a step change in ATP turnover are common not only in yeast but also in cell-free extracts of skeletal muscle and even intact cardiomyocytes. In that light, the observed behaviour is plausible and we can make use of the detailed glycolytic model for explaining the observed behaviour, i.e., caused by a transient mismatch between carbon mass flow through glycolysis and mitochondrial OxPhos.

Action We have rewritten Sections 3.1.1 and 3.1.2 (Results) as well as extended the Discussion on glycolytic oscillations (Section 4.1) to highlight these issues and concerns.

2. Figure 5A is also not consistent with the monoexponential pattern observed during light to moderate exercise intensities. How does this impact the conclusion of the model about H⁺ handling and NADH?

Response: See our response to the previous comment.

Action: –

3. K_{ck} calculation: Golding 1995 (<https://pubmed.ncbi.nlm.nih.gov/7636446/>) provided an equation to K for different conditions of pH and Mg that may better suit the present conditions and improve the accuracy of the model, as K_{ck} can vary 10-fold within a pH range of 6-7.2.

Response: We use the same theoretical framework as Golding et al. (1995) in our model to adjust the equilibrium constants of enzyme-catalyzed reactions to changes in pH and magnesium (cf. Eqn. (9), revised manuscript and Eqn. (9), Golding et al. (1995)). However, in contrast to Golding et al. (1995), in our model the reference equilibrium constants are derived from the free energy of formation of the reference species (see Eqn. (10),

manuscript) and are empirically corrected to the temperature and ionic strength in the model. As the free energy of formation of both HCr and PCr^{2-} was not available, we opted to approximate the apparent equilibrium constant of CK according to Eqn. (13) (original manuscript) with $K^{\text{CK}} = 1.66 \times 10^9 \text{ M}^{-1}$, a parameter value used in multiple previous studies (e.g., Foley et al. (1991), Kushmerick and Meyer (1985)). We note that, although simplified, this approach takes into account changes in pH. Nevertheless, for theoretical self-consistency, we have updated the computation of the CK equilibrium constant in our model (see Section 2.1.2); that is, the revised model uses the same reference equilibrium constant as in Golding et al. (1995). However, we note that our model and Golding et al. (1995) use slightly different binding polynomials. Notably, no major changes are observed in the presented results.

Fig. R1 compares the pH dependency of the apparent CK equilibrium constant in the original model (orange) and the revised model (blue). Given a 10 minute long exercise with an ATPase rate of 30 mM/min and a corresponding drop in pH from about 7.0 to 6.5 (cf. Fig. 5, unrevised manuscript), the original model shows a 3.4-fold increase and the revised model shows a 3.2-fold increase of $K_{\text{app}}^{\text{CK}}$. Even though there is a small shift in absolute values and a deviation of the relative change in $K_{\text{app}}^{\text{CK}}$ over time as a function of pH, the ratio of forward and reverse flux is almost the same with a maximal error of less than 1 %.

Action: We have replaced the simplified pH-dependent description of the CK equilibrium constant by a more detailed description that includes the full binding polynomials. All simulations and figures have been redone accordingly.

4. Page 25: the authors state LDH conversion of pyruvate into lactate has little influence on cellular acid-base balance. I would recommend expanding on this point and adding more results on this specific topic. Is the process of lactate production through glycolysis responsible for muscle acidosis? Can the authors identify some of the key contributors to H^+ production? if pyruvate conversion into lactate is ruled out (cf. present data), what is the contribution of GAPDH or ATP hydrolysis to H^+ balance. The manuscript would benefit from a figure similar to figure 8 with the H^+ production/consumption from GAPDH, ATP hydrolysis and pyruvate conversion into lactate. From the model description, it looks like GAPDH was including in the H^+ balance calculation to run the model.

Response: We thank the reviewer for the constructive feedback regarding cellular acidification. Indeed, we believe that the reviewer's suggestions offer important insights for many readers. Fig. 7 and Fig. A4 in the revised manuscript now present an integrative analysis of myoplasmic pH balance, including coupled ATPase plus resynthesis reactions, whole metabolic pathways, and individual glycolytic enzymes. Notably, in working muscles, protons are tightly coupled, and the net proton generation

flux is small compared to the proton generation flux of individual reactions. The role of individual enzyme reactions is closely related to the next point raised by the reviewer, and a detailed response is given there.

Action: We have added additional data, i.e., Figs. 7 and A4, showcasing the role of different pathways as well as glycolytic enzymes in cellular acidosis.

5. Using their model, can the authors provide insights on how to reconcile the discrepancy between experimental data are consistent with glycolysis contributing to muscle acidosis (<https://pubmed.ncbi.nlm.nih.gov/19066935/>, <https://pubmed.ncbi.nlm.nih.gov/20133437/>), whereas computer models seem to refute this theory (<https://pubmed.ncbi.nlm.nih.gov/20308252/>).

Response: The putative discrepancy between model and experimental observations seems to be a common misunderstanding, caused by diverse (not clearly communicated) perspectives that can be unified using the multi-scale perspective enabled through the proposed model. Many experimental works, e.g., the ones referenced by the reviewer, consider the combined effect of ATPase and ATP resynthesis on the myoplasmic pH balance. Thereby, a net proton uptake is observed for ATPase activity balanced by PCr breakdown and a net proton generation is observed for ATPase flux balanced by glycolytic ATP synthesis. Particularly, the combined ATPase-glycolysis reaction (where each consumed ATP is instantaneously recovered through glycolysis) is often assumed to have a total proton stoichiometry factor of plus two. In working muscle, the composition of the ATP synthesis flux is highly dynamic (see Fig. A4) and thus, in our work as well as in many other modelling studies, ATPase and the ATP recovery processes (CK, glycolysis and OxPhos) are each considered individual pathways. Using this more fine-grained approach, it is observed that the major source of H⁺ production is ATPase. In the glycolytic pathway, the main source of free protons is GAPDH, which is counteracted by the proton consumption of, e.g., LDH and PGK (see Figs. 7 and A4). Overall, glycolysis has a relatively small proton imbalance (which increases over time and with contraction intensity) that contributes to acidification within the cell. The coarse-scale approach, which considers the combined effect of ATPase and glycolysis, is simplified (e.g., neglecting the effect of glycolytic intermediate accumulation) but not inherently wrong. We illustrate this by exemplarily comparing both approaches using the data presented in Marcinek et al. (2010) in Figs. R2 and R3. The model replicates Marcinek et al. (2010) data well, including the glycolytic proton production, implying a one-to-one relationship between glycolytic ATP synthesis and (a fraction of) ATPase. Specifically, the model also shows a (nearly) 1:1 ratio of lactate production and H⁺ generation by the glycolytic pathway in the context of an essentially closed system of (resting) muscle under ischaemia. However, in another setting with oxidative

ATP synthesis through cellular respiration of pyruvate and exercise protocols with accumulation of glycolytic intermediates, a 1:1 ratio of lactate production and H⁺ generation no longer applies even under conditions with stable ATP levels. Thus, we believe that the interpretation that lactate production is responsible for myoplasmic acidification can be misleading, as it pretends that a complex network property is explained by a single observation related to a lumped net reaction.

Action: We have added new Figs. 7 and A4, as well as extended the discussion on myoplasmic pH balance.

Minor comments comments:

1. Introduction page 2: in the context of the skeletal muscle and dynamic muscle contraction, the term fermentation seems inappropriate. I recommend using substrate-level phosphorylation.

Response: Thanks for this suggestion.

Action: We have modified the text accordingly.

Reviewer 2

This paper describes a rigorous computational approach to an important physiological question, The relationship to earlier work is clearly explained, and the limitations are fairly stated. My comments, are largely about further connections which could usefully made to earlier work.

Major comments:

1. **Mitochondrial regulation:** A major simplification is obviously the approximation used for oxidative phosphorylation. As the authors are aware, there is lot of evidence, largely 31P MRS-based, that this relationship holds in general in steady-state aerobic exercise, and in recovery from exercise of a wider range of intensities; it is also consistent with detailed computational-modelling of the whole pathway; and there is quite a large literature on to what extent this relationship is invariant e.g across exercise intensities, and a long debate whether there are or aren't significant additional feed-forward mechanisms. The component reactions are dealt with in some detail in the Appendix, but it's not entirely clear to me how closely the main findings depend on this kinetic approximation, given the importance of this component to overall energy production. This limitation is of course discussed in lines 730-736, although I would say the problem is slightly larger than 'the utilized phenomenological model provides limited insights into the function of mitochondria' - it also has some bearing on at least the robustness of the results about glycolysis.

Response: (1) All kinetic parameters of the phenomenological OxPhos model, i.e. $K_{\text{PYR}}^{\text{OxPhos}}$, $K_{\text{Pi}}^{\text{OxPhos}}$ and the constraints for the optimized parameters $V_{\text{max}}^{\text{OxPhos}}$, $K_{\text{ADP}}^{\text{OxPhos}}$ and n_{H} , have been derived from experimental observations reported in literature. For example, the value of $K_{\text{PYR}}^{\text{OxPhos}}$ is based on the K_{m} value reported for pyruvate translocation into mitochondria Halestrap (1975). (2) The sensitivity analysis did not show any significant influence on the model output given small perturbations of the OxPhos model parameters. In detail, the values from the Kolmogorov-Smirnov statistic ($D_{m,n}$) are smaller than 0.08 for all five OxPhos parameters with respect to the ten selected state variables at all considered work loads. An exception is $V_{\text{max}}^{\text{OxPhos}}$ with slightly higher values ($0.09 \leq D_{m,n} \leq 0.12$) for G3P, PYR and LAC at the high intensity work load. (3) The main limitation of the lumped OxPhos model is that it lacks delayed coupling between demand and supply. This simplification potentially causes an overestimation of the observed glycolytic oscillations. (4) We are excited to add a detailed mitochondria model in the future and how that effect the predictions of the current model.

Action: We acknowledge the the potential overestimation of glycolytic oscillations in the discussion and add the intention of adding a more refined OxPhos model to the outlook.

2. **Overall relationships of key rates and metabolites:** It would be interesting to see both glycolytic and oxidative ATP synthesis rates plotted against the concentrations of the MRS-visible metabolites that have attracted attention as potential closed-loop feedback signals. For oxidative ATP the relationship to [ADP] would obviously be close to the model equation, and the general constraints of the CK equilibrium would no doubt make its relationship to [Pi] near-linear, and that to DeltaG, sigmoid in the expected way (cf Fig 6B, which plots this for total ATP demand). But how does glycolytic ATP synthesis rate look on the same plots? This is not just idle curiosity, as it would throw some useful light on what to expect from 31P MRS data.

In the same spirit, it would be interesting to know see the relationship between total H+ efflux and pH, given the (admittedly inferential) information on this which can be obtained from post-exercise pH recovery kinetics. In this model 'H+ efflux' must be dominated by lactate/H+ transport, a modelled function of intracellular lactate concentration, modified by transmembrane fluxes of bicarbonate (or CO2 fluxes amounting to that). It seems that the Na+/H+ antiporter is not part of the model - what differences does that make? (I think this deserves a comment in the Discussion, at least). I'm aware that this touches on some fundamental aspects of muscle H+ physiology that (even after the apparent resolution of the confusions and misunderstandings detailed in Refs 67-71) are not that well understood. I am just suggesting that the H+ efflux to pH relationship would make an interesting comparison with previous 31P work.

Response: (1) In our model, the feedback regulation of both glycolysis and OxPhos depends on multiple metabolites. Thus, there is no unique relation between ^{31}P -MRS measurable metabolites and the glycolytic or OxPhos-driven ATP resynthesis flux.

(2) Regarding the comment on pH dynamics, we agree with the reviewer that computational modelling is an excellent tool for systematically studying proton balance. We have added new figures, i.e., Fig. 7 and Fig. A4, showing the effect of different metabolic pathways and reactions on proton balance. Particularly considering proton efflux, it is observed that the contribution of the MCT transporter increases over time and with increasing contraction intensity. Although the literature suggests that the role of the Na^+/H^+ antiporter is less critical during exercise, i.e., which is the focus of the proposed manuscript, than during rest, we thank the reviewer for pointing out that this aspect was missing in our discussion.

Action: We have added new figures (Fig. 7 and Fig A4) and extended the discussion on myoplasmic pH balance. Additionally, we addressed the missing Na^+/H^+ antiporter as a limitation of the model under subchapter 4.3.3.

3. **Buffering:** The computational approach taken here is more fundamental and rigorous than what one might call traditional approaches to cytosolic acid base balance. The latter, however, are still useful in quantitative (you might prefer semi-quantitative) analysis of ^{31}P MRS experiments. On a technical level, it would be interesting to know whether the analysis here implies a stoichiometric coefficient for the Lohmann reaction more or less that same as in the detailed analysis of Kushmerick (1997) *Am J Physiol* 272:C1739-47, rather than the larger coefficient of the simpler analysis used in the much of the earlier literature (and still occasionally now).

Response: We thank the reviewer for pointing out that our model potentially can help interpreting ^{31}P -MRS experiments, specifically regarding cytoplasmic proton balance. As mentioned in our response to the comments above, Fig. 7 and Fig. A4 in the revised manuscript now show the role of different metabolic pathways as well as specific glycolytic enzymes on myoplasmic proton balance. We have also added Fig. A5, which summarizes the pH dependency of the proton stoichiometry coefficients. Further, Fig. R4 shows the proton stoichiometry coefficients in the model for three Mg concentrations within the observed concentration range ($1.0\text{ mM} \leq [\text{Mg}^{2+}] \leq 1.2\text{ mM}$) during ATPase rates of up to 30 mM/min . For both ATP hydrolysis and the creatine kinase reaction, the stoichiometry values as functions of pH are indeed very similar to the values reported in the work of Kushmerick (1997).

Action: We have added a new figure to the appendix and addressed the pH dependency of the stoichiometry coefficients in the results and discussion.

4. **Creatin Kinase:** How close is CK to equilibrium through all this? This is a useful technical question for 31P MRS. Is it close enough for the traditional MRS calculation of [ADP] throughout?. The same question could be asked about AMPK and [AMP].

Response: Even with a large CK activity during transitions from rest-to-work or work-to rest at high intensity exercise (ATPase = 30 mM/min) the transient deviation from equilibrium is rather small with a ratio of forward to reverse flux of $0.95 \leq J_{\text{for}}^{\text{CK}}/J_{\text{rev}}^{\text{CK}} \leq 1.1$ (cf. Fig. R5). This aligns well with the simulations presented in Kushmerick (1998). Thus, transient estimates of [ADP] estimated from the CK equilibrium have only a small bias. AMPK (adenosine monophosphate-activated protein kinase) is not considered in our model, but adenylate kinase (AK) is. AK is very close to equilibrium ($J_{\text{for}}^{\text{AK}}/J_{\text{rev}}^{\text{AK}} \approx 1.0$) even throughout high intensity exercise (ATPase = 30 mM/min).

Action: We have added a sentence to the results under subsection 3.2.1.

5. **Off-kinetics:** The fit to post-exercise recovery kinetics of PCr and pH (where the measurement is possible) is quite good. I'm assuming that lactate production rapidly declines after the end of exercise, otherwise pH would keep going down for longer (depending obviously on the balance between this, H+ generation by PCr resynthesis and H+ efflux by various means). I would welcome a bit more discussion of this point. As you allude to, although solid evidence is hard to obtain, the rapid shutdown of lactate production, and the consequent reliance on oxidative ATP synthesis to fund PCr restoration even after significantly 'glycolytic' exercise, is sometimes taken as evidence of some sort of feed-forward control mechanism of the sort that the present analysis shows is not necessary to explain responses during exercise. It would be interesting to see a full time-course (exercise and recovery) showing both glycolytic and oxidative ATP synthesis rate - and also, actually, of lactate efflux and total net H+ efflux, as per my comment above.

Response: The focus of the present manuscript is specifically on reporting our findings regarding the predicted dynamics of myoplasmic energetics, pH, and redox state in the oxidative FT myofiber phenotype during muscular contraction. The dynamic behaviour during recovery is outside the scope of this work. Our choice to include 3 and 6-minute simulations of the latter in Figures 2 and 3, respectively, was specifically to showcase the model performance against typical in vivo experimental data. Figures 5-10 all focus on model predictions of either temporal dynamics or co-variations at the steady-state of selected state variables during simulated contraction. However, we agree with the reviewer that, in the future, it may be of interest to the field, perhaps in particular the in vivo 31P MRS community, to next conduct a study focusing specifically on the dynamics of some of these state variables, including glycolytic and oxidative ATP synthesis rate during post-contractile relaxation .

Action: Though this is an interesting and relevant comment, in favour of limiting the length of the proposed manuscript, we will not add further data and discussion on off-kinetics. However, we add the potential of applying the proposed modelling framework specifically to physiological questions on off-kinetics in the outlook.

Minor comments comments:

1. Fig 6A. I don't quite understand the (so to speak) blanks in the 'CK' block?

Response: At a time frame 10 minutes after the onset of contraction, the contribution of CK to ATP re-synthesis is minimal. Hence, the corresponding area appears to be nearly blank.

Action: We have removed the CK contribution from Figure 6A to avoid confusing the readers.

2. Line 24 - This confounds two things worth distinguishing: the lower ATP yield of glycolysis to lactate, and loss of lactate from the cell.

Response: We recognize the potential for confusion in this sentence and thank the reviewer for pointing it out.

Action: We have rewritten that sentence.

3. Line 175 - not 'acetic' acid.

Response: Thanks for pointing out that mistake.

Action: The sentence now reads "The buffer concentration $[X]_T$ is the sum of the acid $[HX]$ and its conjugate base $[X^-]$ ".

Supporting figures

Figure R1: (A) Apparent CK equilibrium constant in the original implementation (orange) and in the revised manuscript (blue) for an ATPase rate of 30 mM/min. The maximum discrepancy is less than 20 % with respect to the original value. (B) Ratio between the forward and backward flux of the CK reaction. During rest-to-work and work-to-rest transitions, the deviations with respect to the original value are less than 1 %.

Figure R2: Comparison of simulation results (solid lines) and measured ^{31}P -MRS data (circles) collected by Marcinek et al. (2010) for (A) PCr and Pi, (B) ATP (simulation only), (C) pH and (D) cytoplasmic lactate during 25 min of ischemia.

Figure R3: (A) Comparison of calculated glycolytic proton production derived from the ³¹P-MRS data by Marcinek et al. (2010) (circles) and the proton production by the glycolytic pathway in the model (blue line) during 25 min of ischemia. (B) Proton fluxes of the individual pathways in the model (solid lines) and the overall proton flux in the cytoplasm (dashed line). A proton flux indicates proton production and a negative flux indicates proton uptake.

Figure R4: Proton generation stoichiometry coefficient of (A) ATPase and (B) CK as a function of pH. * indicate the stoichiometry coefficients reported by Kushmerick (1997) for [Mg²⁺] = 0.6 mM and [K⁺] = 120 mM. Solid lines indicate the stoichiometry coefficients in our model for three different Mg²⁺ concentrations. [K⁺] is clamped at 120 mM in the model.

Figure R5: Equilibrium state of the creatine kinase (CK) reaction during an explanatory high intensity 10 min exercise protocol at an ATPase rate of 30 mM/min. (A) comparison between K_{app}^{CK} and the reaction quotient Q of CK. (B) ratio of forward and reverse flux.

References

- Foley, J. M., Harkema, S. J., and Meyer, R. A. (1991). Decreased atp cost of isometric contractions in atp-depleted rat fast-twitch muscle. *American Journal of Physiology-Cell Physiology*, 261(5):C872–C881.
- Golding, E. M., Teague Jr, W. E., and Dobson, G. P. (1995). Adjustment of k to varying pH and p_{mg} for the creatine kinase, adenylate kinase and atp hydrolysis equilibria permitting quantitative bioenergetic assessment. *Journal of Experimental Biology*, 198(8):1775–1782.
- Halestrap, A. P. (1975). The mitochondrial pyruvate carrier. kinetics and specificity for substrates and inhibitors. *Biochemical Journal*, 148(1):85–96.
- Kushmerick, M. and Meyer, R. (1985). Chemical changes in rat leg muscle by phosphorus nuclear magnetic resonance. *American Journal of Physiology-Cell Physiology*, 248(5):C542–C549.
- Kushmerick, M. J. (1997). Multiple equilibria of cations with metabolites in muscle bioenergetics. *American Journal of Physiology-Cell Physiology*, 272(5):C1739–C1747.
- Kushmerick, M. J. (1998). Energy balance in muscle activity: simulations of atpase coupled to oxidative phosphorylation and to creatine kinase. *Comparative Biochemistry and Physiology Part B: Biochemistry and Molecular Biology*, 120(1):109–123.
- Marcinek, D. J., Kushmerick, M. J., and Conley, K. E. (2010). Lactic acidosis in vivo: testing the link between lactate generation and H^+ accumulation in ischemic mouse muscle. *Journal of applied physiology*, 108(6):1479–1486.
- Meyer, R. A. (1988). A linear model of muscle respiration explains mono-exponential phosphocreatine changes. *American Journal of Physiology-Cell Physiology*, 254(4):C548–C553.

Dear Dr Klotz,

Re: JP-RP-2025-289702R1 "Dynamic balance of myoplasmic energetics, redox state and protons in a fast-twitch oxidative glycolytic skeletal muscle fiber" by Jana Disch, Jeroen A.L. Jeneson, Daniel A Beard, Oliver Röhrle, and Thomas Klotz

We are pleased to tell you that your paper has been accepted for publication in The Journal of Physiology.

When checking the proof, please make sure that the additional material in the appendix is referred to as the 'appendix' (not supplemental material).

Yours sincerely,

Bettina Mittendorfer
Senior Editor
The Journal of Physiology

IMPORTANT POINTS TO NOTE FOLLOWING ACCEPTANCE OF YOUR PAPER:

- **IMPORTANT NOTICE ABOUT OPEN ACCESS:** To assist authors whose funding agencies mandate immediate public access to published research findings, The Journal of Physiology allows authors to pay an Open Access (OA) fee to have their papers made freely available immediately on publication.

- You can help your research get the attention it deserves! Check out Wiley's free Promotion Guide for best-practice recommendations for promoting your work at: www.wileyauthors.com/eeo/guide. You can learn more about Wiley Editing Services which offers professional video, design, and writing services to create shareable video abstracts, infographics, conference posters, lay summaries, and research news stories for your research at: www.wileyauthors.com/eeo/promotion.

- If you would like to receive our 'Research Roundup', a monthly newsletter highlighting the cutting-edge research published in The Physiological Society's family of journals (The Journal of Physiology, Experimental Physiology, Physiological Reports, The Journal of Nutritional Physiology and The Journal of Precision Medicine: Health and Disease), please click this link, fill in your name and email address and select 'Research Roundup': <https://www.physoc.org/journals-and-media/membernews>

EDITOR COMMENTS

Reviewing Editor:

Thank you for submitting your thorough revisions to The Journal of Physiology. All comments have been adequately addressed, and I would like to congratulate the authors on a very thorough and thought-provoking computational modeling study.

REFEREE COMMENTS

Referee #2:

Thank you - all my comments have been very thoroughly addressed.

Referee #3:

The authors have addressed all my comments.